
# Global symmetries of quantum lattice models under non-invertible dualities

Weiguang Cao[1,2,3]*, Yuan Miao[2]† and Masahito Yamazaki[2,4,5]‡

**1** Center for Quantum Mathematics at IMADA, Southern Denmark University,
Campusvej 55, 5230 Odense, Denmark
**2** Kavli Institute for the Physics and Mathematics of the Universe,
UTIAS, University of Tokyo, Kashiwa, Chiba 277-8583, Japan
**3** Niels Bohr International Academy, Niels Bohr Institute, University of Copenhagen, Denmark
**4** Department of Physics, Graduate School of Science,
University of Tokyo, Tokyo 113-0033, Japan
**5** Trans-Scale Quantum Science Institute, University of Tokyo, Tokyo 113-0033, Japan

★ weiguangcao@imada.sdu.dk , † yuan.miao@ipmu.jp , ‡ masahito.yamazaki@ipmu.jp

## Abstract

Non-invertible dualities/symmetries have become an important tool in the study of quantum field theories and quantum lattice models in recent years. One of the most studied examples is non-invertible dualities obtained by gauging a discrete group. When the physical system has more global symmetries than the gauged symmetry, it has not been thoroughly investigated how those global symmetries transform under non-invertible duality. In this paper, we study the change of global symmetries under non-invertible duality of gauging a discrete group $G$ in the context of (1+1)-dimensional quantum lattice models. We obtain the global symmetries of the dual model by focusing on different Hilbert space sectors determined by the Rep($G$) symmetry. We provide general conjectures of global symmetries of the dual model forming an algebraic ring of the double cosets. We present concrete examples of the XXZ models and the duals, providing strong evidence for the conjectures.

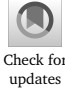

## Contents



# 1 Introduction

Duality is an important concept in theoretical physics. Back in 1941, Kramers and Wannier designed the celebrated duality under their names to study the critical point of the classical Ising model in $2d$ [1]. In recent years, we have gained new understandings of these duality transformations. For example, the Kramers-Wannier (KW) duality now has new interpretation as gauging a 0-form $\mathbb{Z}_2$ symmetry [2,3], a topological defect lines separating two phases [4–8], a non-invertible symmetry in a spin chain [2,3], a sequential quantum circuit [9], and a composition of quantum operations [10]. A general duality transformation can be designed from a given fusion category [7,8,11,12]. Similar to KW duality, some duality transformations are identified as finite group gauging and are non-invertible. These non-invertible duality transformations become global symmetries at self-dual points [2,3], exemplifying the concept of non-invertible symmetry. Even away from the self-dual point, interesting non-invertible symmetries can also be found in the dual theory for some duality transformations [11–15].

    The goal of this paper is to discuss the following questions in the non-invertible duality transformation, in a concrete setup of $(1 + 1)$-dimensional quantum lattice models, where all the operators are defined on a tensorized or constrained Hilbert space that is finite-dimensional locally.[1]

    The first question is how the non-invertible duality transformation affects the symmetries of the theory. While we gauge a group symmetry $G$ under the duality transformation, the symmetry $\mathsf{S}$ of the original theory can be larger than $G$, and the question is to understand the counterpart of the symmetry $\mathsf{S}$ after the gauging. When $\mathsf{S}$ in itself is given by a group, we find that the symmetry in question is given by a ring of the double cosets $G\backslash\mathsf{S}/G$, clarifying imprecise statements in the literature.

    The second question is further refined by considering a detailed analysis of sectors of the theory. We have twist-sectors defined by different twisted boundary conditions, and also symmetry-sectors defined by different representations of the symmetry group. In each sector

---

[1]If we also discretize the time direction, we will arrive at the same setting of statistical mechanics or quantum circuits, such as in [7,8].

we have in general different sector-preserving symmetries, while we will also have symmetries exchanging different sectors. Indeed, the non-invertibility of duality transformation is closely tied to the fact that different sectors are exchanged under the duality transformation. This means that we need to discuss the effect of the duality transformation on each sector, including its effects on the symmetries. Indeed, we can find symmetries acting on the extended Hilbert space, consolidating all the topological sectors. We verify that the familiar statement in the literature that we obtain a symmetry Rep($G$) after gauging $G$, arises in this extended Hilbert space, and the double coset mentioned above arises as one of the ingredients in this discussion relevant to the identity sector of the extended Hilbert space.

Following this idea, we analyze the quantum symmetries after non-invertible duality transformations in the example of XXZ (partially anisotropic Heisenberg) models and their duals. We first perform duality transformations equivalent to gauging a finite Abelian group symmetry ($\mathbb{Z}_2, \mathbb{Z}_3$ in our examples) and find a double coset structure of the dual quantum symmetry. By performing duality transformations sequentially, we can have a composite duality transformation equivalent to gauging a non-Abelian group symmetry ($S_3$) and a Frobenius subalgebra of its representation category (Rep($S_3$)). We also identify duality transformations equivalent to gauging a subalgebra of non-invertible symmetry on the lattice for the first time. We relate our findings to the results from field-theoretical and categorical analysis. We emphasize that to obtain full quantum symmetry in the dual model, we need to turn on twisted boundary conditions and consider twisted duality transformations.

We take a down-to-earth approach with minimal assumptions, where many of our statements can be verified explicitly. While part of our discussion is inspired by works in field theory [16–21] and lattice models [11–13, 22],[2] many of these results are either not necessarily rigorously proven or based on abstract mathematical formalism, and here we present a detailed and down-to-earth discussion of concrete examples, as a step towards more rigorous results on general lattice models.

The rest of this paper is organized as follows. In Sec. 2 we discuss the prototypical example of gauging a $\mathbb{Z}_2$ symmetry of the XXZ model to obtain the Ising zig-zag model. In Sec. 3 we formulate our duality transformation in general and formulate two conjectures. In Sec. 4 we discuss several more examples of the duality transformation between the XXZ model and their duals. We end in Sec. 5 with concluding remarks. We have included several appendices for technical materials.

## 2 Example I: XXZ to Ising zig-zag

Before developing the theory of global symmetries under the duality transformation, we start by explaining an example of gauging the $\mathbb{Z}_2^x$ symmetry of the XXZ model. This example is well-known in the study of symmetry-protected topological orders, in the context of decorated domain-wall [46]. As we shall see, this example contains all the essential ingredients of the general theory in Sec. 3.

### 2.1 XXZ model

We consider the following Hamiltonian defined in the Hilbert space $\mathscr{H}_{\text{XXZ}} = \left(\mathbb{C}^2\right)^{\otimes L}$,

$$\mathbf{H}_{\text{XXZ}} = \sum_{n=1}^{L} c_n \left(X_n X_{n+1} + Y_n Y_{n+1} + \Delta Z_n Z_{n+1}\right), \tag{1}$$

---

[2]While the literature is too vast to be exhausted here, we refer the readers to [11–15, 23–45] for a sample of recent references for non-invertible symmetries in $(1+1)$-dimensional quantum lattice models.

where $\Delta$ is the anisotropic parameter and $X_n$, $Y_n$ and $Z_n$ are the Pauli matrices acting nontrivially on the $n$-th site. We impose a periodic boundary condition $W_{L+1} = W_1$, $W \in \{X, Y, Z\}$; we will also work with twisted (quasi-periodic) boundary conditions later in this paper.

We have introduced the disorder parameters $c_n$. The model is non-integrable for for generic inhomogeneous couplings, except for homogeneous couplings $c_n = c$, $\forall n \in \{1, 2, \ldots, L\}$. Since the global symmetries discussed in the paper are present regardless of disorders, we take $c_n = 1$ for the rest of the paper without losing generality.

**Global symmetries**    The global symmetry of the XXZ model is $O(2) = U(1)^z \rtimes \mathbb{Z}_2^x$. The generator for the $U(1)^z$ symmetry reads

$$\mathcal{O}_Z(\phi) = \prod_{j=1}^{L} \exp\left(i\phi Z_j\right), \quad \phi \in [0, \pi), \tag{2}$$

while the generator for the $\mathbb{Z}_2^x$ symmetry reads

$$\mathcal{O}_X = \prod_{j=1}^{L} X_j. \tag{3}$$

We will hereafter denote the generators of $U(1)^w$ symmetry and $\mathbb{Z}_2^w$ symmetry as

$$\mathcal{O}_W(\phi) := \prod_{j=1}^{L} \exp\left(i\phi W_j\right), \quad \phi \in U(1), \qquad \mathcal{O}_W := \mathcal{O}_W\left(\frac{\pi}{2}\right) = \prod_{j=1}^{L} W_j, \quad W \in \{X, Y, Z\}. \tag{4}$$

An $SU(2)$ symmetry emerges at the isotropic point $\Delta = 1$ of the XXZ model, also known as the XXX (isotropic Heisenberg) model. For an arbitrary group element $g_{\vec{n}} \in SU(2)$, the generator becomes

$$\mathcal{O}_{g_{\vec{n}}} = \exp\left(i\vec{n} \cdot \vec{\sigma}\right), \qquad \vec{n} = (\phi_x, \phi_y, \phi_z), \qquad \vec{\sigma} = (X, Y, Z), \tag{5}$$

where the generators of the $\mathfrak{su}(2)$ algebra are $X = \sum_{j=1}^{L} X_j$, $Y = \sum_{j=1}^{L} Y_j$, and $Z = \sum_{j=1}^{L} Z_j$.

**Twisted boundary conditions**    We consider the XXZ model with twisted (i.e. quasi-periodic) boundary conditions for the latter discussions. In general, the twisted Hamiltonian is given by

$$\mathbf{H}_{\text{XXZ},(e^{i\phi}, W)} = \sum_{j=1}^{L-1}\left(X_j X_{j+1} + Y_j Y_{j+1} + \Delta Z_j Z_{j+1}\right) + e^{-i\phi W_1/2}\left(X_L X_1 + Y_L Y_1 + \Delta Z_L Z_1\right)e^{i\phi W_1/2}, \tag{6}$$

where we included a twist by $W \in \{X, Y, Z\}$ with an angle $\phi$. In this example, we would like to consider the XXZ models with a twist compatible with the $\mathbb{Z}_2^x$ symmetry, i.e. a $\pi$-twist in the $x$ direction. The $\pi$-twisted Hamiltonian reads

$$\mathbf{H}_{\text{XXZ},(-1,X)} = \sum_{j=1}^{L-1}\left(X_j X_{j+1} + Y_j Y_{j+1} + \Delta Z_j Z_{j+1}\right) + \left(X_L X_1 - Y_L Y_1 - \Delta Z_L Z_1\right), \tag{7}$$

defined in the Hilbert space $\mathscr{H}_{\text{XXZ},(-1,X)} \simeq \mathscr{H}_{\text{XXZ}}$. For simplicity of the discussion, we focus on the even-length case for the rest of the paper.

The $(-1, X)$ twisted Hamiltonian (7) no longer possesses the full $O(2)$ symmetry, as the $\pi$ twist explicitly breaks the full $O(2)$ symmetry. The Hamiltonian (7) possesses a $\mathbb{Z}_2^x \times \mathbb{Z}_2^z$ symmetry for even length $L$. When $\Delta = 1$, i.e. at the isotropic point, the $SU(2)$ symmetry of the untwisted Hamiltonian becomes $O(2) = U(1)^x \rtimes \mathbb{Z}_2^z$ instead.

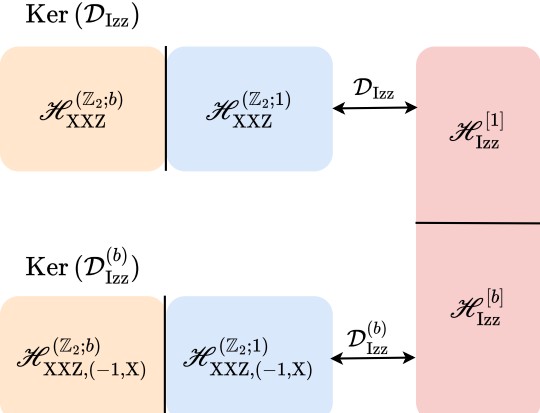

Figure 1: The mapping between the Hilbert spaces of twisted XXZ models and the Izz model. In the Hilbert space of twisted XXZ models, the $\mathbb{Z}_2$ odd sectors (orange) are kernels of the duality operators, while the $\mathbb{Z}_2$ even sectors (blue) are mapped to the Hilbert space of untwisted Izz model (red). We use the same notation in figures of other examples later in Sec. 4.

## 2.2 Gauging the $\mathbb{Z}_2$ symmetry: XXZ to Ising zig-zag

We will now gauge the $\mathbb{Z}_2^x$ symmetry of the XXZ (XXX) models (with and without twist compatible with the $\mathbb{Z}_2^x$ symmetry), resulting in the model called Ising zig-zag model [13] (also known as the dual XXZ model in the literature [47–50]). We use the following notation for the $\mathbb{Z}_2$ group that contains two elements $\{1, b\}$ with $b^2 = 1$.

**Duality operator** By gauging the $\mathbb{Z}_2^x$ symmetry, we transform the XXZ model to the Ising zig-zag (Izz) model. The duality operator is related to the renowned Kramers–Wannier duality operator (dressed by an additional unitary transformation), which can be written as a matrix product operator (MPO) with bond dimension $\chi = 2$ [3, 7, 13],

$$\mathcal{D}_{\text{Izz}} = \text{Tr}_a \left( \prod_{n=1}^{L} \mathbf{A}_{a,n} \right), \qquad \mathbf{A}_{a,n} = \begin{pmatrix} |\uparrow\rangle\langle+|_n & -|\uparrow\rangle\langle-|_n \\ |\downarrow\rangle\langle-|_n & -|\downarrow\rangle\langle+|_n \end{pmatrix}_a, \tag{8}$$

where $|\uparrow\rangle, |\downarrow\rangle$ and $|+\rangle, |-\rangle$ are eigenstates with $\pm 1$ eigenvalues of the Pauli $Z$ and $X$ operators, respectively. The auxiliary space $a$ is 2-dimensional, i.e. the bond dimension is 2.

The relation between the duality operator $\mathcal{D}_{\text{Izz}}$ and gauging the $\mathbb{Z}_2^x$ symmetry can be understood as inserting dynamical $\mathbb{Z}_2$ gauge fields between each lattice sites and integrating out the original spins. This will result in the Izz Hamiltonian, which is equivalent to applying the duality operator to the XXZ model Hamiltonian. Consequently, $\mathcal{D}_{\text{Izz}}$ projects the states from the Hilbert space $\mathscr{H}_{\text{XXZ}}$ to the $\mathbb{Z}_2$ invariant subspace [2, 3, 25]. Therefore, we shall not distinguish between the gauging of a discrete symmetry and the corresponding duality operator.

The Hamiltonian of the Izz model reads

$$\mathbf{H}_{\text{Izz}} = \sum_{j=1}^{L} \left( Z_{j-1} Z_{j+1} (1 + X_j) - \Delta X_j \right), \tag{9}$$

which is related to the XXZ Hamiltonian via the duality operator,

$$\mathcal{D}_{\text{Izz}} \cdot \mathbf{H}_{\text{XXZ}} = \mathbf{H}_{\text{Izz}} \cdot \mathcal{D}_{\text{Izz}}. \tag{10}$$

The dual Hilbert space $\mathscr{H}_{\text{Izz}} = \left(\mathbb{C}^2\right)^{\otimes L}$ happens to be isomorphic to $\mathscr{H}_{\text{XXZ}}$. This is not the case for the other examples in Sec. 4.

**Twist sectors**   To take into account the entire Hilbert space of $\mathbf{H}_{\text{Izz}}$, we need to consider the XXZ model with a $(-1, X)$ twist (7). After a duality transformation in the presence of the twist, we have

$$\mathcal{D}_{\text{Izz}}^{(b)} \cdot \mathbf{H}_{\text{XXZ},(-1,X)} = \mathbf{H}_{\text{Izz}} \cdot \mathcal{D}_{\text{Izz}}^{(b)}, \tag{11}$$

where

$$\mathcal{D}_{\text{Izz}}^{(b)} = Z_1 \mathcal{D}_{\text{Izz}}^{(1)}, \tag{12}$$

where $\mathcal{D}_{\text{Izz}}^{(1)} = \mathcal{D}_{\text{Izz}}$ and the superscripts are labeled by the $\mathbb{Z}_2$ elements 1 and $b$.

The duality operators satisfy

$$\mathcal{D}_{\text{Izz}}^{\dagger} \mathcal{D}_{\text{Izz}} = \left(\mathcal{D}_{\text{Izz}}^{(b)}\right)^{\dagger} \cdot \mathcal{D}_{\text{Izz}}^{(b)} = \mathbb{I} + \prod_{n=1}^{L} X_n, \tag{13}$$

where the right-hand side is proportional to a projector to the even sector of the $\mathbb{Z}_2^x$ symmetry of the XXZ model (with and without twist) that we gauge. These relations can be explicitly deduced from the MPO formalism of the duality operators (8).

The Izz Hamiltonian possesses $\text{Rep}(\mathbb{Z}_2) \simeq \mathbb{Z}_2$ symmetry as a consequence of gauging an Abelian group symmetry $\mathbb{Z}_2$, generated by

$$(-1)^L \prod_{n=1}^{L} X_n. \tag{14}$$

Meanwhile, we have the following properties of the twisted duality operators,

$$\mathcal{D}_{\text{Izz}}^{(\sigma)} \cdot \left(\mathcal{D}_{\text{Izz}}^{(\sigma)}\right)^{\dagger} = \begin{cases} \mathbb{I} + (-1)^L \prod_{n=1}^{L} X_n, & \sigma = 1, \\ \mathbb{I} - (-1)^L \prod_{n=1}^{L} X_n, & \sigma = b. \end{cases} \tag{15}$$

It implies that $\mathcal{D}_{\text{Izz}}^{(\sigma)} \cdot \left(\mathcal{D}_{\text{Izz}}^{(\sigma)}\right)^{\dagger}$ are projectors, projecting onto two orthogonal parts of the Hilbert space $\mathscr{H}_{\text{Izz}}$.[3]

**Symmetry sectors**   Due to Maschke's theorem [51], the Hilbert space of the Izz Hamiltonian can be labeled by the conjugacy class of $\mathbb{Z}_2$[4]

$$\mathscr{H}_{\text{Izz}} = \bigoplus_{g \in \mathbb{Z}_2} \mathscr{H}_{\text{Izz}}^{[g]} = \mathscr{H}_{\text{Izz}}^{[1]} \oplus \mathscr{H}_{\text{Izz}}^{[b]}, \tag{16}$$

where the superscript stands for the eigenvalues of (14). The Hilbert spaces of the XXZ models factorize into

$$\mathscr{H}_{\text{XXZ}} = \bigoplus_{\hat{g} \in \text{Rep}(\mathbb{Z}_2)} \mathscr{H}_{\text{XXZ}}^{(\mathbb{Z}_2;\hat{g})} = \mathscr{H}_{\text{XXZ}}^{(\mathbb{Z}_2;1)} \oplus \mathscr{H}_{\text{XXZ}}^{(\mathbb{Z}_2;b)},$$
$$\mathscr{H}_{\text{XXZ},(-1,X)} = \bigoplus_{\hat{g} \in \text{Rep}(\mathbb{Z}_2)} \mathscr{H}_{\text{XXZ},(-1,X)}^{(\mathbb{Z}_2;\hat{g})} = \mathscr{H}_{\text{XXZ},(-1,X)}^{(\mathbb{Z}_2;1)} \oplus \mathscr{H}_{\text{XXZ},(-1,X)}^{(\mathbb{Z}_2;b)}. \tag{17}$$

---

[3]The entire Hilbert space of the Izz model $\mathscr{H}_{\text{Izz}} \simeq \mathscr{H}_{\text{XXZ}}^{(\mathbb{Z}_2;1)} \oplus \mathscr{H}_{\text{XXZ},(-1,X)}^{(\mathbb{Z}_2;1)}$, which can be considered as a subspace of the extended Hilbert space of the XXZ models before gauging.

[4]Since group $\mathbb{Z}_2$ is Abelian, the conjugacy class of $\mathbb{Z}_2$ is isomorphic to itself, where $[1] = \{1\}$ and $[b] = \{b\}$.

From (13) and (15), we can deduce that, the duality operators act non-trivially on the two sectors of Hilbert space $\mathcal{H}_{\text{Izz}}$, i.e.

$$\mathcal{D}_{\text{Izz}} \in \text{Hom}\left(\mathcal{H}_{\text{XXZ}}^{(\mathbb{Z}_2;1)}, \mathcal{H}_{\text{Izz}}^{[1]}\right), \qquad \mathcal{D}_{\text{Izz}}^{(b)} \in \text{Hom}\left(\mathcal{H}_{\text{XXZ},(-1,X)}^{(\mathbb{Z}_2;1)}, \mathcal{H}_{\text{Izz}}^{[b]}\right). \tag{18}$$

Moreover, considering the operator acting in the entire Hilbert space $\mathcal{H}_{\text{Izz}}$, we have

$$\sum_{\sigma \in \mathbb{Z}_2} \mathcal{D}_{\text{Izz}}^{(\sigma)} \cdot \left(\mathcal{D}_{\text{Izz}}^{(\sigma)}\right)^{\dagger} = |G|\, \mathbb{I}_{\text{Izz}}, \tag{19}$$

where the order of the group $G = \mathbb{Z}_2$ is $|G| = 2$.

**Symmetries.** We gauge the $\mathbb{Z}_2^x$ group, which is a non-normal subgroup of $O(2)$. We start with the even sector of the Hilbert space of $\mathbf{H}_{\text{Izz}}$. We arrive at the ring of double cosets $\mathbb{Z}[\mathbb{Z}_2^x \backslash O(2)/ \mathbb{Z}_2^x]$, with the generators of the previously $U(1)^z$ part

$$\mathcal{O}_z^{\text{Izz}}(\phi) = \mathcal{D}_{\text{Izz}}\left(\prod_{n=1}^{L} \exp\left(i\phi Z_n\right)\right) \mathcal{D}_{\text{Izz}}^{\dagger}, \tag{20}$$

where the group multiplication is modified into

$$\mathcal{O}_z^{\text{Izz}}(\phi_1) \cdot \mathcal{O}_z^{\text{Izz}}(\phi_2) = \left(\mathcal{O}_z^{\text{Izz}}(\phi_1 + \phi_2) + \mathcal{O}_z^{\text{Izz}}(\phi_1 - \phi_2)\right), \tag{21}$$

in accord with the multiplication of the ring of double cosets. The detailed explanation will be given in Sec. 3 and App. B.

This symmetry is referred to as cosine symmetry in [23]. In fact, the multiplication rule can be derived either using the properties of the duality operators (13) and (15), or expressing the operator $\mathcal{O}_z^{\text{Izz}}(\phi)$ as an MPO from the explicit MPO formalism of the operator $\mathcal{O}_z^{\text{Izz}}(\phi)$, which we present in the App. C. The cosine symmetry (in the critical regime $|\Delta| \leq 1$) is a reminiscence of the $\mathbb{Z}_2$ orbifolded $U(1)$ symmetry in the $c = 1$ compactified free boson CFT [13].

The cosine symmetry has been previously identified with a coset $O(2)/\mathbb{Z}_2^x$ in [23]. We will show in Sec. 3 and App. B that this identification is, in fact, inappropriate. The mathematical structure of the cosine symmetry is an algebraic ring with elements being the double cosets $\mathbb{Z}_2 \backslash O(2)/\mathbb{Z}_2$.[5] The addition and multiplication of the ring are defined in App. B.

From the discussions above, we know that the cosine symmetry acts only non-trivially in the Hilbert space $\mathcal{H}_{\text{Izz}}^{[1]}$. For the other half of the Hilbert space $\mathcal{H}_{\text{Izz}}^{[b]}$, $\frac{\mathbb{Z}_2 \times \mathbb{Z}_2}{\mathbb{Z}_2} = \mathbb{Z}_2$ symmetry remains after gauging, where the generator of the remaining $\mathbb{Z}_2$ symmetry in $\mathcal{H}_{\text{Izz}}^{[b]}$ sector is

$$\mathcal{D}_{\text{Izz}}^{(b)} \cdot \prod_{j=1}^{L} Z_j \cdot \left(\mathcal{D}_{\text{Izz}}^{(b)}\right)^{\dagger} = \left(\mathbb{I} - (-1)^L \prod_{j=1}^{L} X_j\right) \cdot \prod_{m=1}^{L/2} X_{2m-1}. \tag{22}$$

It is easy to check that $\prod_{m=1}^{L/2} X_{2m-1}$ is a global symmetry in $\mathcal{H}_{\text{Izz}}^{[1]}$ sector too. Therefore, for the entire Hilbert space $\mathcal{H}_{\text{Izz}}$, the global symmetry acting on the entire Hilbert space becomes $\mathbb{Z}_2 \times \mathbb{Z}_2$, generated by the operators $(-1)^L \prod_{j=1}^{L} X_j$ and $\prod_{m=1}^{L/2} X_{2m-1}$, while the sector $\mathcal{H}_{\text{Izz}}^{[1]}$ has the enriched cosine symmetry.

At the isotropic point, according to the derivation outlined in Sec. 3, we expect the Izz model with $\Delta = 1$ to have the $\mathbb{Z}[\mathbb{Z}_2 \backslash SU(2)/\mathbb{Z}_2]$ symmetry in Hilbert space sector $\mathcal{H}_{\text{Izz}}^{[1]}$, and $O(2)^* = O(2)'/\mathbb{Z}_2^x$ symmetry[6] in Hilbert space sector $\mathcal{H}_{\text{Izz}}^{[b]}$, where the symmetry of (7) with

---

[5]The appearance of the double coset was noted in [52] and Example 9.7.4 of [53].

[6]$\mathbb{Z}_2^x$ is a normal subgroup of $O(2)' = U(1)^x \rtimes \mathbb{Z}_2^z$. We use $O(2)'$ and $O(2)^*$ (both are $O(2)$ symmetries) to distinguish from the $O(2) = U(1)^z \rtimes \mathbb{Z}_2^x$ symmetry of the XXZ model without twist.

$\Delta = 1$ is $O(2)' = U(1)^x \rtimes \mathbb{Z}_2^z$. (The $O(2) \subset \mathbb{Z}[\mathbb{Z}_2\backslash SU(2)/\mathbb{Z}_2]$ symmetry acts on both sectors of the Hilbert space $\mathcal{H}_{\mathrm{Izz}}^{[1]}$ and $\mathcal{H}_{\mathrm{Izz}}^{[b]}$, thus a global symmetry of the entire Hilbert space $\mathcal{H}_{\mathrm{Izz}}$.)

Even though the ring of double cosets $\mathbb{Z}[\mathbb{Z}_2\backslash SU(2)/\mathbb{Z}_2]$ does not have a group structure, it still contains a $U(1)$ part, i.e.

$$\mathcal{D}_{\mathrm{Izz}}\left(\prod_{j=1}^{L}\exp(\mathrm{i}\phi X_j)\right)\mathcal{D}_{\mathrm{Izz}}^{\dagger} \cdot \mathcal{D}_{\mathrm{Izz}}\left(\prod_{j=1}^{L}\exp(\mathrm{i}\theta X_j)\right)\mathcal{D}_{\mathrm{Izz}}^{\dagger} = \mathcal{D}_{\mathrm{Izz}}\left(\prod_{j=1}^{L}\exp(\mathrm{i}(\phi+\theta)X_j)\right)\mathcal{D}_{\mathrm{Izz}}^{\dagger}. \quad (23)$$

If we combine the two $U(1)$ symmetries in each sector, we obtain a $U(1)$ symmetry that acts on the entire Hilbert space $\mathcal{H}_{\mathrm{Izz}}$,

$$\begin{aligned}
\mathcal{O}_{U(1)}^{\mathrm{Izz}}(\phi) &= \frac{1}{2}\left(\mathcal{D}_{\mathrm{Izz}}\left(\prod_{j=1}^{L}\exp(\mathrm{i}\phi X_j)\right)\mathcal{D}_{\mathrm{Izz}}^{\dagger} + \mathcal{D}_{\mathrm{Izz}}^{(b)}\left(\prod_{j=1}^{L}\exp(\mathrm{i}\phi X_j)\right)\left(\mathcal{D}_{\mathrm{Izz}}^{(b)}\right)^{\dagger}\right) \\
&= \prod_{j=1}^{L}\exp(\mathrm{i}\phi Z_j Z_{j+1}).
\end{aligned} \quad (24)$$

The generator of the $U(1)$ symmetry counts the number of domain walls in the x-direction, i.e.

$$\sum_{j=1}^{L}Z_j Z_{j+1}. \quad (25)$$

Combining the $\mathbb{Z}_2$ symmetry $\prod_{m=1}^{L/2}X_{2m-1}$ and the U(1) symmetry (25), we obtain the $O(2)$ symmetry acting on the entire Hilbert space $\mathcal{H}_{\mathrm{Izz}}$. Meanwhile, we still need to take into account the Rep($\mathbb{Z}_2$) $\simeq \mathbb{Z}_2$ symmetry generated by $(-1)^L\prod_{j=1}^{L}X_j$ that commute with the $O(2)$ generators, which result in a global symmetry $O(2)\times\mathbb{Z}_2$. In the meantime, the cosine symmetry is still present, as part of the ring of double cosets $\mathbb{Z}[\mathbb{Z}_2\backslash SU(2)/\mathbb{Z}_2]$.

**Relation to Rep($S_3$) and Rep($D_8$) symmetries.** When we specify the cosine symmetry parameter $\phi$ to root-of-unity values ($e^{\mathrm{i}\phi N} = 1$ with $N \in \mathbb{N}$), we can obtain other categorical symmetries that are related to the category of representations of dihedral groups $D_{2n}$.

Here we give the explicit expressions for the Rep($S_3$) ($n = 3$, $D_6 \simeq S_3$) and Rep($D_8$) ($n = 4$) symmetries.

Consider

$$\mathcal{O}_z^{\mathrm{Izz}}(0) = \mathbb{I} + (-1)^L\prod_{n=1}^{L}X_n = \mathbb{I} + \mathbf{R}, \quad (26)$$

where operator $\mathbf{R}$ is the $\mathbb{Z}_2$ symmetry of the Izz model. Moreover, we consider the following relations,

$$\mathcal{O}_z^{\mathrm{Izz}}\left(\frac{2\pi}{3}\right) \cdot \mathcal{O}_z^{\mathrm{Izz}}\left(\frac{2\pi}{3}\right) = \mathcal{O}_z^{\mathrm{Izz}}(0) + \mathcal{O}_z^{\mathrm{Izz}}\left(\frac{4\pi}{3}\right) = \mathbb{I} + \mathbf{R} + \mathcal{O}_z^{\mathrm{Izz}}\left(\frac{2\pi}{3}\right), \quad (27)$$

and

$$\mathbf{R} \cdot \mathcal{O}_z^{\mathrm{Izz}}\left(\frac{2\pi}{3}\right) = \mathcal{O}_z^{\mathrm{Izz}}\left(\frac{2\pi}{3}\right). \quad (28)$$

We can then identify the operator

$$\mathbf{S} = \mathcal{O}_z^{\mathrm{Izz}}\left(\frac{2\pi}{3}\right), \quad (29)$$

such that

$$\mathbf{S}^2 = \mathbb{I} + \mathbf{R} + \mathbf{S}, \qquad \mathbf{R} \cdot \mathbf{S} = \mathbf{S}. \quad (30)$$

The relations above coincide with those of the fusion algebra of the Rep($S_3$) category: the operators $\{\mathbb{I}, \mathbf{R}, \mathbf{S}\}$ give a representation of the Rep($S_3$) category $\{1, r, s\}$ on the tensorized Hilbert space $\left(\mathbb{C}^2\right)^L$ with fusion algebra[7]

$$r^2 = 1, \qquad r \cdot s = s, \qquad s^2 = 1 + r + s. \tag{31}$$

This fact has been discussed in [12, 13].

When the system size $L$ is even, we can find a realization of the Rep($D_8$) category from the cosine symmetry. We have an additional $\mathbb{Z}_2 \times \mathbb{Z}_2$ symmetry at even $L$, which can be seen from the operator

$$\mathcal{O}_z^{\text{Izz}}\left(\frac{\pi}{2}\right) = \mathbf{R}_o + \mathbf{R}_e, \tag{32}$$

where two $\mathbb{Z}_2$ operators are

$$\mathbf{R}_o = \prod_{n=1}^{L/2} X_{2n-1}, \qquad \mathbf{R}_e = \prod_{n=1}^{L/2} X_{2n}. \tag{33}$$

Together with operators mentioned previously $\mathbb{I}$ and $\mathbf{R} = \mathbf{R}_o \cdot \mathbf{R}_e$, they form a representation of the Klein 4-group $\mathbb{Z}_2 \times \mathbb{Z}_2$.

Moreover, we consider the following relations,

$$\mathcal{O}_z^{\text{Izz}}\left(\frac{\pi}{4}\right) \cdot \mathcal{O}_z^{\text{Izz}}\left(\frac{\pi}{4}\right) = \mathcal{O}_z^{\text{Izz}}(0) \cdot \mathcal{O}_z^{\text{Izz}}\left(\frac{\pi}{2}\right) = \mathbb{I} + \mathbf{R}_o + \mathbf{R}_e + \mathbf{R}, \tag{34}$$

$$\mathbf{R}_o \cdot \mathcal{O}_z^{\text{Izz}}\left(\frac{\pi}{4}\right) = \mathbf{R}_e \cdot \mathcal{O}_z^{\text{Izz}}\left(\frac{\pi}{4}\right) = \mathbf{R} \cdot \mathcal{O}_z^{\text{Izz}}\left(\frac{\pi}{4}\right) = \mathcal{O}_z^{\text{Izz}}\left(\frac{\pi}{4}\right). \tag{35}$$

When we identify the non-invertible object $\mathbf{T} = \mathcal{O}_z^{\text{Izz}}(\pi/4)$, and the relations reproduce the fusion algebra of the Rep($D_8$) category:[8] $\{\mathbb{I}, \mathbf{R}_o, \mathbf{R}_e, \mathbf{R}, \mathbf{T}\}$ is the representation of the Rep($D_8$) category $\{1, r_o, r_e, r, t\}$ on the tensorized Hilbert space $\left(\mathbb{C}^2\right)^L$.[9]

## 3 A general theory of symmetries under dualities

In the following, we outline the properties of the dual Hamiltonian after gauging a group symmetry $G$, and how the global symmetry of the Hamiltonian $\mathsf{S} \supset G$ transforms under a gauging (i.e. duality transformation).

### 3.1 Gauging a group

**Generalities on gauging** We start with a Hamiltonian $\mathbf{H}$ defined on a Hilbert space $\mathcal{H}$. We assume that the Hamiltonian $\mathbf{H}$ has a global symmetry $\mathsf{S}$, i.e.

$$[\mathbf{H}, \mathcal{O}_s] = 0, \quad s \in \mathsf{S}, \tag{36}$$

where $\mathcal{O}_s$ is the representation of $\mathsf{S}$ on the Hilbert space $\mathcal{H}$. The symmetry $\mathsf{S}$ is described mathematically by a monoidal category.

---

[7]The three irreducible representations are: 1 is the trivial representation, $r$ is the sign representation, and $s$ is the standard two-dimensional representation.

[8]Actually the fusion algebra is the same for category $TY(\mathbb{Z}_2 \times \mathbb{Z}_2)$ with different bicharacters and Frobenius-Schur indicators.

[9]The irreducible representations of $D_8$ are: the trivial representation 1, three sign representations $r_o, r_e, r$, and a two-dimensional representations $t$ where $a$ is given by a $\pi/2$ rotation and $b$ is a reflection.

We consider the gauging of a discrete symmetry $S'$ of $\mathbf{H}$, where $S'$ is a fusion subcategory of $S$. After gauging we obtain the dual Hamiltonian $\tilde{\mathbf{H}}$ defined in a different Hilbert space $\tilde{\mathcal{H}}$ via the duality operator $\mathcal{D} \in \mathrm{Hom}(\mathcal{H}, \tilde{\mathcal{H}})$, such that $\mathcal{D}$ intertwines $\mathbf{H}$ and $\tilde{\mathbf{H}}$:

$$\mathcal{D} \cdot \mathbf{H} = \tilde{\mathbf{H}} \cdot \mathcal{D}. \tag{37}$$

For the lattice models we consider, $S'$ is either a finite group $G$ or its category of representations $\mathrm{Rep}(G)$; for the former, a model can be realized by specifying a fusion category $\mathcal{C} = \mathrm{Rep}(G)$ and the corresponding module category $\mathcal{M}$. The duality operator $\mathcal{D}$ is thus the bimodule functor between two module categories $\mathcal{M}$ and $\mathcal{N}$ [11, 12], which can be viewed as a domain wall in the field theory picture [54].

We focus on the strongly symmetric duality,[10] which requires further properties on the duality operator $\mathcal{D}$. The duality operators that carry out the gauging procedure are assumed to be *strongly symmetric* [55], i.e.

$$\mathcal{D} \cdot \mathcal{O}_{s'} = \dim(s')\mathcal{D}, \quad s' \in S', \tag{38}$$

where $\mathcal{O}_{s'} \in \mathrm{End}(\mathcal{H})$ is the representation of the symmetry $S'$ in the Hilbert space $\mathcal{H}$. Quantum dimension is $\dim(s') = 1$ if $S'$ is a group $G$, which needs not be the case for a general fusion category.

In general, the duality operator can be weakly symmetric [55], especially when gauging a generic categorical symmetry.[11] We do not extend the discussions to the most general case and postpone the categorical study of the relations between the global symmetries and weakly symmetric dualities to future works.

A consequence of gauging is that

$$\mathcal{D}^\dagger \cdot \mathcal{D} = \sum_{s' \in S'} \dim(s') \mathcal{O}_{s'} \in \mathrm{End}(\mathcal{H}), \tag{39}$$

where $\mathcal{D}^\dagger \in \mathrm{Hom}(\tilde{\mathcal{H}}, \mathcal{H})$ is an operator representing the gauging of the symmetry dual to $S'$. The right-hand side of this equation is a symmetric Frobenius algebra object of category $\mathcal{C}(G)$ (the category canonically associated with the group $G$), which gives a general defining data of gauging [17, 56, 57]. This is compatible with the strongly symmetric condition (38), as shown in App. A.

**Gauging a group**    Let us now specialize in the case where $S'$ is a group $G$, which is either Abelian or non-Abelian.

Let us consider the twisted Hamiltonian $\mathbf{H}^g$ with $g \in G$. We denote the corresponding Hilbert space by $\mathcal{H}_g$, and the symmetry of the twisted Hamiltonian by $S_g \subset S$. If $G$ is non-Abelian, the symmetry $G$ of the original theory is broken to the stabilizer (commutant subgroup):

$$G_g := \{k \in G \,|\, k \cdot g = g \cdot k\} \subset G. \tag{40}$$

Consequently, we can only gauge the subgroup $G_g \subset S_g$ symmetry for the twisted Hamiltonian.

From Maschke's theorem [51], the Hilbert space can be decomposed into

$$\mathcal{H} = \bigoplus_{\hat{g} \in \mathrm{Rep}(G)} \mathcal{H}^{\hat{g}}, \tag{41}$$

---

[10]The gauging of finite discrete groups will lead to strongly symmetric duality operators [55].

[11]The gauging of a Frobenius subalgebra of a fusion category might lead to weakly symmetric duality operator. We demonstrate this in the example of Sec. 4.3.

with respect to the symmetry group $G$. The dual Hilbert space is decomposed into the conjugacy class $\mathrm{Conj}(G)$ of $G$ [58]:

$$\tilde{\mathcal{H}} = \bigoplus_{c \in \mathrm{Conj}(G)} \tilde{\mathcal{H}}^c \,, \tag{42}$$

where we define

$$\mathrm{Conj}(G) := \{[g], g \in G\}\,, \qquad [g] := \{h \cdot g \cdot h^{-1}, h \in G\}\,. \tag{43}$$

For Abelian groups, each element belongs to its own conjugacy class. The identity element 1 always belongs to the conjugacy class $[1]$.

This is due to the Tannaka–Krein duality, which establishes exact mappings between the category of representations $\mathrm{Rep}(G)$ and the conjugacy class $\mathrm{Conj}(G)$ [58]. A simple observation is that the number of conjugacy classes of $G$ is equal to the number of irreducible representations, which is the number of simple objects in $\mathrm{Rep}(G)$.

When we apply (39) to a group $G$, we obtain

$$\mathcal{D}^\dagger \cdot \mathcal{D} = \sum_{g \in G} \mathcal{O}_g \in \mathrm{End}(\mathcal{H})\,. \tag{44}$$

The dual object is in the dual category $\mathrm{Rep}(G)$, and by exchanging the role of $G$ and $\mathrm{Rep}(G)$ in (39), we obtain

$$\mathcal{D} \cdot \mathcal{D}^\dagger = \sum_{\hat{g} \in \mathrm{Rep}(G)} \dim(\hat{g})\, \tilde{\mathcal{O}}_{\hat{g}} \in \mathrm{End}(\tilde{\mathcal{H}})\,, \tag{45}$$

where $\dim(\hat{g})$ is the quantum dimension of the simple object $\hat{g}$ in category $\mathrm{Rep}(G)$.

**Duality transformations on twist/symmetry sectors**    In order to find symmetries in all sectors, we need to move on to the dualities between the twisted Hamiltonian and the dual Hamiltonian.

We gauge the discrete symmetry $G_g$ of the twisted Hamiltonian $\mathbf{H}^g$, resulting in the same dual Hamiltonian through duality operator $\mathcal{D}^{(g)}$,

$$\mathcal{D}^{(g)} \cdot \mathbf{H}^g = \tilde{\mathbf{H}} \cdot \mathcal{D}^{(g)}\,. \tag{46}$$

Since we gauge the discrete symmetry $G_g$, we expect the duality operator to satisfy

$$\left(\mathcal{D}^{(g)}\right)^\dagger \cdot \mathcal{D}^{(g)} = \sum_{k \in G_g} \mathcal{O}_k \in \mathrm{End}\left(\mathcal{H}_g\right)\,, \tag{47}$$

where the formula corresponds to the symmetric Frobenius algebra object of the discrete subgroup $G_g$ [17].

We conjecture that the dual formula to be

$$\mathcal{D}^{(g)} \cdot \left(\mathcal{D}^{(g)}\right)^\dagger = \sum_{\hat{g} \in \mathrm{Rep}(G)} \chi_{\hat{g}}(g)\, \tilde{\mathcal{O}}_{\hat{g}} \in \mathrm{End}\left(\tilde{\mathcal{H}}\right)\,, \tag{48}$$

where $\chi_{\hat{g}}(g) = \mathrm{Tr}[\hat{g}(g)]$ is the character for the $\hat{g}$ evaluated at $g$.

For $g = 1$ we have $\chi_{\hat{g}}(1) = \dim(\hat{g})$, which recover the previous formula (45). The conjecture implies that for $g$ and $h$ in the same conjugacy class,

$$\mathcal{D}^{(g)} \cdot \left(\mathcal{D}^{(g)}\right)^\dagger = \mathcal{D}^{(h)} \cdot \left(\mathcal{D}^{(h)}\right)^\dagger\,, \qquad [g] = [h]\,. \tag{49}$$

The conjecture also implies

$$\sum_{g \in G} \mathcal{D}^{(g)} \cdot \left(\mathcal{D}^{(g)}\right)^\dagger = \sum_{g \in G} \sum_{\hat{g} \in \mathrm{Rep}(G)} \chi_{\hat{g}}(g) \tilde{\mathcal{O}}_{\hat{g}} = |G| \tilde{\mathcal{O}}_{\hat{1}} = |G|, \tag{50}$$

where we used the orthogonality of the character.

Another conjecture is about the reconstruction of the dual Hilbert space.

For the twisted Hamiltonian, the Hilbert space is decomposed with respect to the category $\mathrm{Rep}(G_g)$, i.e.

$$\mathscr{H}_g = \bigoplus_{\hat{g} \in \mathrm{Rep}(G_g)} \mathscr{H}_g^{\hat{g}}. \tag{51}$$

In addition, we conjecture that

$$\tilde{\mathscr{H}}^c \simeq \mathscr{H}_g^{\hat{1}}, \quad \text{when} \quad g \in c. \tag{52}$$

From (47) and (48), we realize that both $(\mathcal{D}^{(g)})^\dagger \mathcal{D}^{(g)}$ and $\mathcal{D}^{(g)}(\mathcal{D}^{(g)})^\dagger$ are proportional to a projector in Hilbert spaces $\mathscr{H}_g$ and $\tilde{\mathscr{H}}$. We therefore conjecture that the kernel of the duality operator (with or without twist) is precisely

$$\ker \mathcal{D}^{(g)} = \mathscr{H} / \mathscr{H}_g^{\hat{1}}, \quad \forall g \in c, \tag{53}$$

and analogously the cokernel of the duality operator becomes

$$\mathrm{coker} \, \mathcal{D}^{(g)} = \ker \left(\mathcal{D}^{(g)}\right)^\dagger = \tilde{\mathscr{H}} / \tilde{\mathscr{H}}^c, \quad \forall g \in c. \tag{54}$$

It is straightforward to show that (53) and (54) are compatible with the properties of duality operators (47) and (48).

Hence, the isomorphism between the aforementioned Hilbert sub-spaces can be realized through duality operators,

$$\frac{1}{\sqrt{|G_g|}} \mathcal{D}^{(g)}: \quad \mathscr{H}_g^{\hat{1}} \to \tilde{\mathscr{H}}^c, \tag{55}$$

$$\frac{1}{\sqrt{|G_g|}} \left(\mathcal{D}^{(g)}\right)^\dagger: \quad \tilde{\mathscr{H}}^c \to \mathscr{H}_g^{\hat{1}}, \tag{56}$$

so that

$$\frac{1}{|G_g|} \left(\mathcal{D}^{(g)}\right)^\dagger \mathcal{D}^{(g)}: \quad \mathscr{H}_g^{\hat{1}} \to \mathscr{H}_g^{\hat{1}}, \tag{57}$$

$$\frac{1}{|G_g|} \mathcal{D}^{(g)} \left(\mathcal{D}^{(g)}\right)^\dagger: \quad \tilde{\mathscr{H}}^c \to \tilde{\mathscr{H}}^c, \tag{58}$$

are identity operators in the Hilbert sub-spaces $\mathscr{H}_g^{\hat{1}}$ and $\tilde{\mathscr{H}}^c$, respectively.

As a consequence of the conjectures, from (37) and (46), we can relate the spectra of the (twisted) Hamiltonian and the dual Hamiltonian, i.e.

$$\mathrm{Spec}_{\mathbf{H}}\left(\mathscr{H}^{\hat{1}}\right) = \mathrm{Spec}_{\tilde{\mathbf{H}}}\left(\tilde{\mathscr{H}}^{[1]}\right), \tag{59}$$

$$\mathrm{Spec}_{\mathbf{H}_g}\left(\mathscr{H}_g^{\hat{1}}\right) = \mathrm{Spec}_{\tilde{\mathbf{H}}}\left(\tilde{\mathscr{H}}^c\right), \quad g \in c. \tag{60}$$

The entire spectrum of the dual Hamiltonian thus can be obtained by studying parts of the spectra of twisted Hamiltonian,

$$\mathrm{Spec}_{\tilde{\mathbf{H}}}\left(\tilde{\mathscr{H}}\right) = \bigcup_{x=1}^{\ell} \mathrm{Spec}_{\mathbf{H}_{g_x}}\left(\mathscr{H}_{g_x}^{\hat{1}}\right), \quad g_x \in c_x, \tag{61}$$

where $c_1, c_2, \ldots, c_\ell$ are the $\ell$-many conjugacy classes of the group $G$.

## 3.2 Symmetries under duality

We now move on to the original question of how global symmetries transform under the duality (gauging) transformation.

To answer this question, we use the "sandwiched construction" of the symmetry operators, keeping in mind that they only act non-trivially on Hilbert sub-spaces according to the decomposition.

The duality operator can be considered as a "half-gauging" operation [16]. By acting the symmetry operator of the original Hamiltonian between two duality operators (half-gaugings), such an operator naturally commutes with the dual Hamiltonian. We refer to this procedure as the sandwiched construction. In terms of formulae, we start with a Hamiltonian $\mathbf{H}$ with group symmetry $\mathsf{S}$. There is a discrete sub-symmetry $G \subset \mathsf{S}$, which we will gauge. The resulting Hamiltonian is $\tilde{\mathbf{H}}$, with at least $\mathrm{Rep}(G)$ symmetry as a result of gauging $G$. We will answer the question of what happens to the full symmetry $\mathsf{S}$.

The gauging is through a strongly symmetric duality operator $\mathcal{D} \in \mathrm{Hom}(\mathcal{H}, \tilde{\mathcal{H}})$, i.e.

$$\mathcal{D} \cdot \mathbf{H} = \tilde{\mathbf{H}} \cdot \mathcal{D} \,. \tag{62}$$

Let us start with the $\mathsf{S}$ symmetry operator $\mathcal{O}_s \in \mathrm{End}(\mathcal{H})$, for generators $s \in \mathsf{S}$. It is straightforward to observe that

$$\tilde{\mathcal{O}}_s = \mathcal{D} \cdot \mathcal{O}_s \cdot \mathcal{D}^\dagger \in \mathrm{End}(\tilde{\mathcal{H}}) \tag{63}$$

commutes with the dual Hamiltonian $\tilde{\mathbf{H}}$,

$$\left[\tilde{\mathcal{O}}_s, \tilde{\mathbf{H}}\right] = 0 \,, \quad s \in \mathsf{S} \,. \tag{64}$$

The operators $\tilde{O}_s$ form a representation of a ring of double cosets $\tilde{\mathsf{S}} = \mathbb{Z}[G\backslash\mathsf{S}/G]$,[12] as demonstrated in App. B. In the context of category theory, a similar discussion of the role of the ring of double cosets in the mathematical literature was mentioned in [52].

However, $\tilde{\mathsf{S}}$ only acts non-trivially on the Hilbert sub-space $\tilde{\mathcal{H}}^{[1]} \simeq \mathcal{H}^{\hat{1}}$. The symmetries in the cokernel of $\mathcal{D}$ are not included. Therefore, we need to consider the twisted Hamiltonians $\mathbf{H}^g$.

For the twisted Hamiltonians $\mathbf{H}^g$, the total symmetries satisfy $\mathsf{S}_g \subset \mathsf{S}$. The discrete symmetries are the stabilizer subgroup $G_g \subset G$, which we can gauge. We have the sandwiched symmetry operator of $\tilde{\mathbf{H}}$,

$$\tilde{\mathcal{O}}_s^g = \mathcal{D}^{(g)} \cdot \mathcal{O}_s^g \cdot \left(\mathcal{D}^{(g)}\right)^\dagger \,, \quad s \in \mathsf{S}_g \,, \tag{65}$$

which again form a representation of the ring of double cosets $\tilde{\mathsf{S}}_g = \mathbb{Z}[G_g\backslash\mathsf{S}_g/G_g]$.

**Global symmetries of the entire dual Hilbert space**   From the sandwiched construction, we are able to obtain global symmetries of the dual Hamiltonian that act on certain sectors of the Hilbert space according to the $\mathrm{Rep}(G)$ symmetry. What remains to be addressed are the global symmetries that the Hilbert space $\tilde{\mathcal{H}}$ has.

Combining the global symmetries obtained via the sandwiched construction, we obtain the common part between symmetries of different sectors, i.e.

$$\tilde{\mathsf{S}}_{\mathrm{comm}} = \bigcap_{g \in G} \tilde{\mathsf{S}}_g \,, \tag{66}$$

where the generators act on the entire dual Hilbert space $\tilde{\mathcal{H}}$ by adding different contributions,

$$\tilde{\mathcal{O}}_s = \sum_{g \in G} \frac{1}{|G_g|} \mathcal{D}^{(g)} \mathcal{O}_s \left(\mathcal{D}^{(g)}\right)^\dagger \,, \quad s \in \mathsf{S}_{\mathrm{comm}} \,, \tag{67}$$

---

[12] When $G$ is a normal subgroup of $\mathsf{S}$, the ring of double cosets becomes the quotient group $\mathsf{S}/G$.

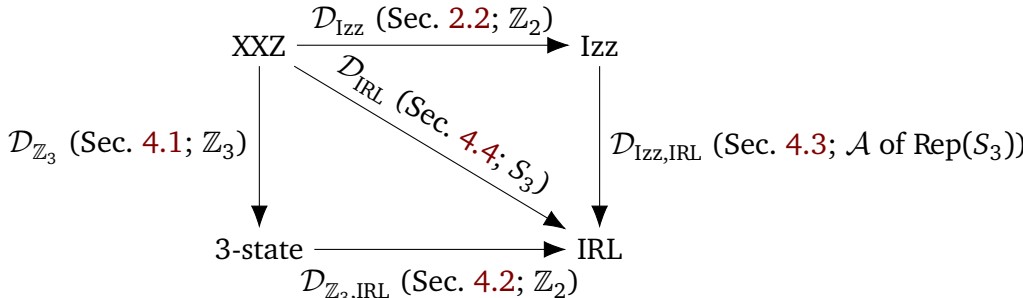

Figure 2: Four models with the dualities between them, where the symmetries gauged are shown inside the brackets. The arrow from the Izz model to the IRL model represents a gauging of the Frobenius subalgebra $\mathcal{A}$ of the fusion-category $\mathrm{Rep}(S_3)$ symmetry, while other lines represent gaugings of finite discrete groups. The diagonal arrow from the XXZ model to the IRL model requires the gauging of the non-Abelian $S_3$ symmetry.

where the common part of the global symmetries of each twisted Hamiltonian is

$$S_{\mathrm{comm}} = \bigcap_{g \in G} S_g. \tag{68}$$

If the group $G$ is Abelian, the common part of the global symmetries in the dual Hilbert space can be expressed as

$$\tilde{S}_{\mathrm{comm}} = \mathbb{Z}[G \backslash S_{\mathrm{comm}} / G]. \tag{69}$$

Moreover, the dual Hamiltonian has $\mathrm{Rep}(G)$ symmetry, which leads to a decomposition of the Hilbert space. Since the symmetry $\tilde{S}_{\mathrm{comm}}$ acts in the dual Hilbert space $\tilde{\mathcal{H}}$, all the generators of $\tilde{S}_{\mathrm{comm}}$ commute with the generators of $\mathrm{Rep}(G)$. Therefore, we have the global symmetry of the entire dual Hilbert space of the dual Hamiltonian $\tilde{\mathcal{H}}$ as

$$\tilde{S}_{\mathrm{comm}} \times \mathrm{Rep}(G). \tag{70}$$

We remark that the dual Hamiltonian might have additional global symmetries other than (70), which do not commute with the $\mathrm{Rep}(G)$ symmetry. The reason is that the generators of the additional symmetry do not just act within each sector from the decomposition according to the $\mathrm{Rep}(G)$ symmetry. Those additional symmetries cannot be obtained directly from the sandwiched construction, and we will present such an example in Sec. 4.1.

# 4 XXZ models and their duals

We write down the general theory of global symmetries under the gauging of a discrete group, which is associated with strongly symmetric dualities in Sec. 3. In the following, we will demonstrate the general theory above using a few concrete examples with the gauging of spin-1/2 XXZ models[13] and their duals. Those examples provide evidence for the conjectures in Sec. 3.

We will start in Sec. 4.1 by gauging the Abelian group $\mathbb{Z}_3$, which results in the 3-state antiferromagnet model. The 3-state antiferromagnet model also has the $O(2)$ symmetry of the XXZ models, where the $U(1)$ part can be deduced from the sandwiched construction in Sec. 3.

---

[13] Even though the XXZ model is integrable, integrability does not play any significant role in the discussion of global symmetries.

Table 1: Global symmetries of the XXZ and XXX Hamiltonians with different twists. We assume the system sizes to be even.

| Hamiltonian | Global symmetries | Hamiltonian | Global symmetries |
|---|---|---|---|
| $\mathbf{H}_{\text{XXZ}}$ | $O(2) = U(1)^z \rtimes \mathbb{Z}_2^x$ | $\mathbf{H}_{\text{XXX}}$ | $SU(2)$ |
| $\mathbf{H}_{\text{XXZ},(-1,X)}$ | $\mathbb{Z}_2^x \times \mathbb{Z}_2^z$ | $\mathbf{H}_{\text{XXX},(-1,X)}$ | $O(2) = U(1)^x \rtimes \mathbb{Z}_2^z$ |
| $\mathbf{H}_{\text{XXZ},(\omega,Z)}$ | $U(1)^z$ | $\mathbf{H}_{\text{XXX},(\omega,Z)}$ | $U(1)^z$ |
| $\mathbf{H}_{\text{XXZ},(\omega^2,Z)}$ | $U(1)^z$ | $\mathbf{H}_{\text{XXX},(\omega^2,Z)}$ | $U(1)^z$ |

Table 2: Symmetries of different models when $|\Delta| \neq 1$ with even length.

| Model | Gauged symmetry from XXZ | Global symmetries by sectors | Global symmetries in the entire dual Hilbert space |
|---|---|---|---|
| XXZ | | | $O(2)$ |
| Izz | $\mathbb{Z}_2$ | $\mathbb{Z}[\mathbb{Z}_2 \backslash O(2)/\mathbb{Z}_2] \mid \mathbb{Z}_2$ | $\mathbb{Z}_2 \times \mathbb{Z}_2$ |
| 3-state | $\mathbb{Z}_3$ | $O(2) \mid U(1) \mid U(1)$ | $(U(1) \times \mathbb{Z}_3) \rtimes \mathbb{Z}_2$ |
| IRL | $S_3$ | $\mathbb{Z}[\mathbb{Z}_2 \backslash O(2)/\mathbb{Z}_2] \mid U(1) \mid \mathbb{Z}_2$ | $\mathbb{Z}_2 \times \text{Rep}(S_3)$ |

Table 3: Symmetries of different models at isotropic point $\Delta = 1$ with even length. The $O(2)^*$ symmetry is the $O(2)$ symmetry coming from the ungauged $U(1)^x \rtimes \mathbb{Z}_2^z$ symmetry of the XXX model with $(-1, X)$ twist.

| Model | Gauged symmetry from XXX | Global symmetries by sectors | Global symmetries in the entire dual Hilbert space |
|---|---|---|---|
| XXX | | | $SU(2)$ |
| Izz | $\mathbb{Z}_2$ | $\mathbb{Z}[\mathbb{Z}_2 \backslash SU(2)/\mathbb{Z}_2] \mid O(2)^*$ | $O(2)^* \times \mathbb{Z}_2$ |
| 3-state | $\mathbb{Z}_3$ | $\mathbb{Z}[\mathbb{Z}_3 \backslash SU(2)/\mathbb{Z}_3] \mid U(1) \mid U(1)$ | $(U(1) \times \mathbb{Z}_3) \rtimes \mathbb{Z}_2$ |
| IRL | $S_3$ | $\mathbb{Z}[S_3 \backslash SU(2)/S_3] \mid U(1) \mid O(2)^*$ | $\mathbb{Z}_2 \times \text{Rep}(S_3)$ |

We then continue in Sec. 4.2 with gauging the $\mathbb{Z}_2$ symmetry of the 3-state antiferromagnet model, leading to a Rydberg ladder model. Similarly, we obtain the same Rydberg ladder model by gauging the Frobenius subalgebra $\mathcal{A}$ of the $\text{Rep}(S_3)$ categorical symmetry of the Izz model. As discussed in Sec. 4.3, the duality operator in this example is weakly symmetric, whose details are postponed to future investigation. However, the exact form of the duality operator is useful for our study of gauging the non-Abelian discrete group symmetry.

Finally, by combining the two gaugings, we study in Sec. 4.2 the gauging of the non-Abelian $S_3$ symmetry of the XXZ models. The decomposition of Hilbert spaces in the presence of twists and the properties of the duality operators match perfectly with the conjectures in Sec. 3, providing solid evidence for the latter. We believe that the examples can be easily generalized to other instances.

The results of the global symmetries in all three models are curated in Tables 2 and 3.

## 4.1 XXZ to 3-state antiferromagnet model

By gauging the $\mathbb{Z}_3^z$ symmetry, we transform the XXZ model to the 3-state antiferromagnet (AFM) model, where the $\mathbb{Z}_3$ group is $\{1, a, a^2\}$ with $a^3 = 1$.

**Duality operators and twist sectors**    The duality operator can be written in terms of a bond-dimension-3 MPO,

$$\mathcal{D}_{\mathbb{Z}_3} = \text{Tr}_a \left( \prod_{n=1}^{L} \mathbf{B}_{a,n} \right), \qquad \mathbf{B}_{a,n} = \begin{pmatrix} 0 & |1\rangle\langle\uparrow|_n & |2\rangle\langle\downarrow|_n \\ |0\rangle\langle\downarrow|_n & 0 & |2\rangle\langle\uparrow|_n \\ |0\rangle\langle\uparrow|_n & |1\rangle\langle\downarrow|_n & 0 \end{pmatrix}_a, \qquad (71)$$

which is a generalization of the $\mathbb{Z}_2$ gauging case.

The Hilbert space of the 3-state AFM is a constrained Hilbert space, $\mathcal{H}_{\mathbb{Z}_3} \subset \left(\mathbb{C}^3\right)^L$, where the state $|s_1, s_2, \ldots, s_L\rangle$ satisfies

$$s_j \neq s_{j+1}, \qquad s_j \in \{0, 1, 2\}, \quad j \in \{1, 2, \ldots, L\}, \tag{72}$$

with periodic boundary condition $s_{L+1} = s_1$. The dimension of the Hilbert space is $2^L + (-1)^L$ [13].

The Hamiltonian of the 3-state AFM reads

$$\mathbf{H}_{\mathbb{Z}_3} = \sum_{m=1}^{L} \mathcal{P}_{\mathbb{Z}_3} \left[ \sum_{s=0}^{2} \mathbf{E}_{m-1}^{s,s} \mathbf{E}_{m+1}^{s,s} \left(\mathbf{E}_m^{s-1,s+1} + \mathbf{E}_m^{s+1,s-1}\right) + \Delta\left(\mathbf{E}_{m-1}^{s,s}\mathbf{E}_{m+1}^{s+1,s+1} + \mathbf{E}_{m-1}^{s,s}\mathbf{E}_{m+1}^{s-1,s-1} - \mathbb{I}_{\mathbb{Z}_3}\right) \right] \mathcal{P}_{\mathbb{Z}_3}^\dagger, \tag{73}$$

where $\mathcal{P}_{\mathbb{Z}_3} \cdots \mathcal{P}_{\mathbb{Z}_3}^\dagger$ projects the Hilbert space $\left(\mathbb{C}^3\right)^L$ to $\mathcal{H}_{\mathbb{Z}_3}$, cf. (72) and the local operator $\mathbf{E}_m^{a,b} = \mathbb{I}_3^{\otimes(m-1)} \otimes (|a\rangle\langle b|) \otimes \mathbb{I}_3^{\otimes(L-m)}$. The identity operator $\mathbb{I}_3$ acts on $\mathbb{C}^3$ and $\mathbb{I}_{\mathbb{Z}_3}$ is the identity operator of $\mathcal{H}_{\mathbb{Z}_3}$. Physically, the first part of (73) means that if $(m-1)$-th and $(m+1)$-th sites are both in $|s\rangle$ state, the Hamiltonian flips the $m$-th site from $(s \pm 1)$ to $(s \mp 1)$, while the second part of (73) gives a potential proportional to $\Delta$ when $(m-1)$-th and $(m+1)$-th sites are in different state. Another version of the Hamiltonian in terms of the 3-state Potts operators can be found in Eq. (3.14) of [13].

Locally the duality maps the state in the Hilbert space of the XXZ model to the state in the Hilbert space $\mathcal{H}_{\mathbb{Z}_3}$ [13]

$$\mathcal{D}_{\mathbb{Z}_3} : |\cdots\uparrow\cdots\rangle \to |\cdots s, (s+1)\cdots\rangle, \quad |\cdots\downarrow\cdots\rangle \to |\cdots s, (s-1)\cdots\rangle, \quad s \in \{0, 1, 2\}, \tag{74}$$

due to the property of the MPO structure (71).

Since we gauge an Abelian $\mathbb{Z}_3$ symmetry, we can take into account the twisted Hamiltonian. In this case, we need to consider the XXZ model with a $2s\pi/3$ twist in the z-axis, i.e.

$$\mathbf{H}_{\text{XXZ}, (\omega^s, Z)} = \sum_{j=1}^{L-1} \left(X_j X_{j+1} + Y_j Y_{j+1} + \Delta Z_j Z_{j+1}\right) + 2\left(\omega^s \sigma_L^+ \sigma_1^- + \omega^{-s} \sigma_L^- \sigma_1^+\right) + \Delta Z_L Z_1, \tag{75}$$

where $\omega = e^{2i\pi/3}$ is the third root of unity. Here we denote $\sigma_j^\pm = (X_j \pm iY_j)/2$.

Meanwhile, the duality operator of the twisted sectors can be expressed as,

$$\mathcal{D}_{\mathbb{Z}_3}^{(a^x)} = \left[ \sum_{s=0}^{2} \omega^{-xs}\left(\mathbf{E}_L^{s,s}\mathbf{E}_1^{s+1,s+1} + \mathbf{E}_L^{s,s}\mathbf{E}_1^{s-1,s-1}\right) \right] \mathcal{D}_{\mathbb{Z}_3}, \quad x \in \{0, 1, 2\}. \tag{76}$$

The duality between the twisted Hamiltonians and the dual Hamiltonian becomes

$$\mathcal{D}_{\mathbb{Z}_3}^{(a^s)} \cdot \mathbf{H}_{\text{XXZ}, (\omega^s, Z)} = \mathbf{H}_{\mathbb{Z}_3} \cdot \mathcal{D}_{\mathbb{Z}_3}^{(a^s)}, \quad s \in \{0, 1, 2\}. \tag{77}$$

Moreover, the duality operators satisfy the following properties,

$$\left(\mathcal{D}_{\mathbb{Z}_3}^{(a^s)}\right)^\dagger \cdot \mathcal{D}_{\mathbb{Z}_3}^{(a^s)} = \mathbb{I} + \prod_{n=1}^{L} \exp\left(\frac{2i\pi}{3} Z_n\right) + \prod_{n=1}^{L} \exp\left(\frac{4i\pi}{3} Z_n\right), \tag{78}$$

denoting the $\mathbb{Z}_3^z$ symmetry of the XXZ models (with and without twists) that we gauge.

$$\mathcal{D}_{\mathbb{Z}_3}^{(a^s)} \cdot \left(\mathcal{D}_{\mathbb{Z}_3}^{(a^s)}\right)^\dagger = \mathbb{I}_{\mathbb{Z}_3} + \omega^s \mathsf{X}_{\mathbb{Z}_3} + \omega^{-s} \mathsf{X}_{\mathbb{Z}_3}^2, \quad s \in \{0, 1, 2\}, \tag{79}$$

where the $\text{Rep}(\mathbb{Z}_3) \simeq \mathbb{Z}_3$ symmetry of the 3-state AFM model becomes

$$\mathsf{X}_{\mathbb{Z}_3} = \mathcal{P}_{\mathbb{Z}_3} \cdot \mathsf{X} \cdot \mathcal{P}_{\mathbb{Z}_3}^\dagger = \mathcal{P}_{\mathbb{Z}_3} \cdot \left( \prod_{n=1}^{L} \begin{pmatrix} 0 & 1 & 0 \\ 0 & 0 & 1 \\ 1 & 0 & 0 \end{pmatrix}_n \right) \cdot \mathcal{P}_{\mathbb{Z}_3}^\dagger. \tag{80}$$

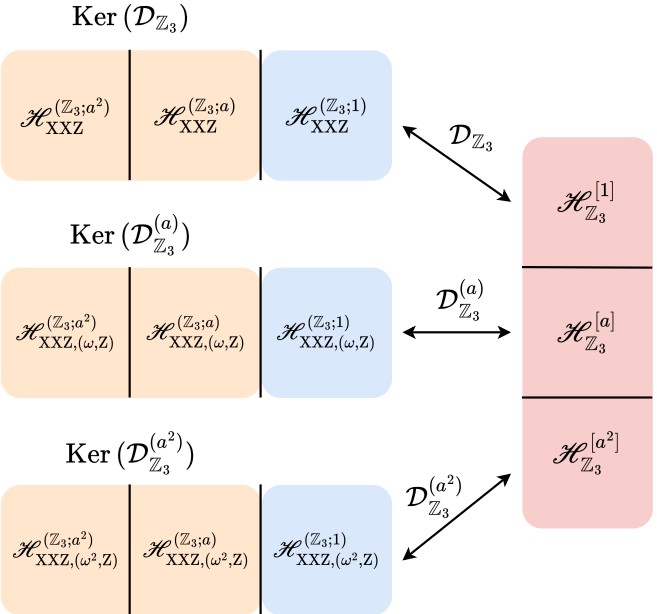

Figure 3: The mapping between the Hilbert spaces of twisted XXZ models and the 3-state AFM model.

**Symmetries**  For the twisted Hamiltonian (75), the symmetry becomes $U(1)^z$, instead of the full $O(2)$ symmetry or $SU(2)$ symmetry at the isotropic point.

The symmetry of the 3-state AFM model in the sector $\mathscr{H}_{\mathbb{Z}_3}^{[1]}$ is the quotient group $\frac{O(2)}{\mathbb{Z}_3^z} \simeq O(2)$, since $\mathbb{Z}_3^z$ is a normal subgroup of the $O(2)$ symmetry. Moreover, for the twisted sectors, $\mathsf{S}_a = \mathsf{S}_{a^2} = U(1)^z$, and the resulting symmetries are the quotient group $\frac{U(1)}{\mathbb{Z}_3^z} \simeq U(1)$ in each sectors.

Combining the $U(1)$ part in all three sectors, we obtain a $U(1)$ symmetry that act on the dual Hilbert space $\mathscr{H}_{\mathbb{Z}_3}$, i.e.

$$\mathcal{O}_{U(1)}^{\mathbb{Z}_3}(\phi) = \frac{1}{3}\sum_{s=0}^{2}\left[\left(\mathcal{D}_{\mathbb{Z}_3}^{(a^s)}\right)^\dagger \mathcal{O}_z(\phi)\mathcal{D}_{\mathbb{Z}_3}^{(a^s)}\right] = \prod_{n=1}^{L}\exp\left(i\phi\sum_{s=0}^{2}\left(\mathbf{E}_n^{s,s}\mathbf{E}_{n+1}^{s+1,s+1} - \mathbf{E}_n^{s,s}\mathbf{E}_{n+1}^{s-1,s-1}\right)\right). \quad (81)$$

The generator of the $U(1)$ symmetry is thus

$$\mathsf{W} = \sum_{n=1}^{L}\sum_{s=0}^{2}\left(\mathbf{E}_n^{s,s}\mathbf{E}_{n+1}^{s+1,s+1} - \mathbf{E}_n^{s,s}\mathbf{E}_{n+1}^{s-1,s-1}\right), \quad (82)$$

which counts the number of plus and minus domain walls. The plus/minus domain walls are defined as $|\ldots,s,s\pm 1,\ldots\rangle$ respectively.

In the $\mathscr{H}_{\mathbb{Z}_3}^{[1]}$ sector, the additional $\mathbb{Z}_2$ symmetry (part of the $O(2)$ symmetry) reads

$$\frac{1}{3}\mathcal{D}_{\mathbb{Z}_3}\prod_{n=1}^{L}X_n\mathcal{D}_{\mathbb{Z}_3}^\dagger = \frac{1}{3}\left(\mathbb{I}_{\mathbb{Z}_3} + \mathsf{X}_{\mathbb{Z}_3} + \mathsf{X}_{\mathbb{Z}_3}^2\right)\mathsf{F}_{\mathbb{Z}_3}, \quad (83)$$

where the $\mathbb{Z}_2$ operator is

$$\mathsf{F}_{\mathbb{Z}_3} = \mathcal{P}_{\mathbb{Z}_3}\prod_{n=1}^{L}\begin{pmatrix} 1 & 0 & 0 \\ 0 & 0 & 1 \\ 0 & 1 & 0 \end{pmatrix}_n \mathcal{P}_{\mathbb{Z}_3}^\dagger, \quad (84)$$

satisfying $\mathsf{F}_{\mathbb{Z}_3}^2 = \mathbb{I}_{\mathbb{Z}_3}$.

The $\mathbb{Z}_2$ operator (84) flips all the plus domain walls to minus domain walls and vice versa, similar to the spin-flip operator in the XXZ model. In addition to the sector $\mathscr{H}_{\mathbb{Z}_3}^{[1]}$, the $\mathbb{Z}_2$ operator (84) acting on the sector $\mathscr{H}_{\mathbb{Z}_3}^{[a]}$ ($\mathscr{H}_{\mathbb{Z}_3}^{[a^2]}$) is not a symmetry but an *isomorphism* from $\mathscr{H}_{\mathbb{Z}_3}^{[a]}$ ($\mathscr{H}_{\mathbb{Z}_3}^{[a^2]}$) to $\mathscr{H}_{\mathbb{Z}_3}^{[a^2]}$ ($\mathscr{H}_{\mathbb{Z}_3}^{[a]}$). This means that the $\mathbb{Z}_2$ operator (84) is also a symmetry of the dual Hilbert space $\mathscr{H}_{\mathbb{Z}_3}$, i.e.

$$\left[ \mathsf{F}_{\mathbb{Z}_3}, \mathbf{H}_{\mathbb{Z}_3} \right] = 0\,. \tag{85}$$

Physically, the $\mathbb{Z}_2$ symmetry of flipping domain walls is a reminiscence of the $\mathbb{Z}_2^x$ symmetry of the XXZ model in the $\mathscr{H}_{\mathbb{Z}_3}^{[1]}$ sector, and maps states to $\mathscr{H}_{\mathbb{Z}_3}^{[a]}$ to $\mathscr{H}_{\mathbb{Z}_3}^{[a^2]}$ and vice versa. This is an extra symmetry, due to the fact that the two twisted Hamiltonians share the same spectra. We should regard this $\mathbb{Z}_2$ symmetry as an additional symmetry beyond the sandwiched construction. Together with the $U(1)$ part, we have the full $O(2) = U(1) \rtimes \mathbb{Z}_2$ symmetry of the dual Hilbert space $\mathscr{H}_{\mathbb{Z}_3}$.

Meanwhile, the $\mathrm{Rep}(\mathbb{Z}_3)$ symmetry generated by $\mathsf{X}_{\mathbb{Z}_3}$ commute with the $U(1)$ part and anticommute with the $\mathbb{Z}_2$ part of the global symmetry, resulting in the final answer to the global symmetry of $\mathbf{H}_{\mathbb{Z}_3}$, i.e.

$$(U(1) \times \mathbb{Z}_3) \rtimes \mathbb{Z}_2\,. \tag{86}$$

In the isotropic case, the $SU(2)$ symmetry becomes the ring of double cosets $\mathbb{Z}[\mathbb{Z}_3^z \backslash SU(2)/ \mathbb{Z}_3^z]$ in the Hilbert sub-space $\mathscr{H}_{\mathbb{Z}_3}^{[1]}$, which contains a $U(1)$ symmetry

$$\frac{1}{3} \mathcal{D}_{\mathbb{Z}_3}^\dagger \cdot \mathcal{O}_z(\phi) \cdot \mathcal{D}_{\mathbb{Z}_3}\,. \tag{87}$$

For a generic $SU(2)$ generator $\mathcal{O}_{g_{\vec{n}}}$ that is parametrized by $\vec{n} = (\phi_x, \phi_y, \phi_z)$, after the sandwiching procedure, we have

$$\mathcal{O}_{g_{\vec{n}}}^{\mathbb{Z}_3} = \mathcal{D}_{\mathbb{Z}_3}^\dagger \cdot \mathcal{O}_{g_{\vec{n}}} \cdot \mathcal{D}_{\mathbb{Z}_3}\,, \tag{88}$$

with the fusion product

$$\mathcal{O}_{g_{\vec{n}}}^{\mathbb{Z}_3} \cdot \mathcal{O}_{g_{\vec{m}}}^{\mathbb{Z}_3} = \mathcal{O}_{g_{\vec{n}} \cdot g_{\vec{m}}}^{\mathbb{Z}_3} + \mathcal{O}_{g_{\vec{n}} \cdot g_{(0,0,\frac{2\pi}{3})} \cdot g_{\vec{m}}}^{\mathbb{Z}_3} + \mathcal{O}_{g_{\vec{n}} \cdot g_{(0,0,-\frac{2\pi}{3})} \cdot g_{\vec{m}}}^{\mathbb{Z}_3}\,. \tag{89}$$

At the isotropic point, however, we do not recover the $SU(2)$ symmetry of the XXX model. Instead, we have the same global symmetry acting on the dual Hilbert space $\mathscr{H}_{\mathbb{Z}_3}$, $(U(1) \times \mathbb{Z}_3) \rtimes \mathbb{Z}_2$. In the $\mathscr{H}_{\mathbb{Z}_3}^{[1]}$ sector, additional symmetry of $\mathbb{Z}[\mathbb{Z}_3 \backslash SU(2)/\mathbb{Z}_3]$ is present at the isotropic point compared to the $O(2)$ symmetry with $|\Delta| \neq 1$.

## 4.2 3-state antiferromagnet to integrable Rydberg ladder

If we further gauge the $\mathbb{Z}_2$ symmetry (84) of the 3-state AFM model, we will arrive at the integrable Rydberg ladder (IRL) model [12,13,22]. The physical realization of the IRL model using Rydberg atoms can be found in [13,22]. We again use $\{1, b\}$ to denote the group elements of $\mathbb{Z}_2$.

The Hilbert space of the IRL model is a different constrained Hilbert space, $\mathscr{H}_{\mathrm{IRL}} \subset \left( \mathbb{C}^3 \right)^L$ with the rule $|\{h_1, h_2, \dots\}\rangle$ such that

$$h_j = 0 \text{ or } h_j = 2 \ \Rightarrow \ h_{j-1} = h_{j+1} = 1; \quad h_j = 1 \ \Rightarrow \ h_{j-1} \in \{0, 1, 2\}, \quad h_{j+1} \in \{0, 1, 2\}\,. \tag{90}$$

The IRL Hamiltonian is written as

$$
\mathbf{H}_{\mathrm{IRL}} = \sum_{m=1}^{L} \mathcal{P}_{\mathrm{IRL}} \Bigg[ \sum_{h,k \in \{0,1\}} (1-2h)(1-2k) \mathbf{E}_{m-1}^{2h,2h} \mathbf{E}_{m}^{1,1} \mathbf{E}_{m+1}^{2k,2k} + \sum_{h \in \{0,1\}} \frac{1}{\sqrt{2}} \mathbf{E}_{m-1}^{1,1} \mathbf{E}_{m+1}^{1,1} \left( \mathbf{E}_{m}^{1,2h} + \mathbf{E}_{m}^{2h,1} \right)
$$

$$
+ \Delta \left( \mathbf{E}_{m-1}^{2h,2h} \mathbf{E}_{m}^{1,1} \mathbf{E}_{m+1}^{1,1} + \mathbf{E}_{m-1}^{1,1} \mathbf{E}_{m}^{1,1} \mathbf{E}_{m+1}^{2h,2h} \right) + \frac{1}{2} \mathbf{E}_{m-1}^{1,1} \left( \mathbf{E}_{m}^{2h,2h} + \mathbf{E}_{m}^{2h,2-2h} \mathbf{E}_{m+1}^{2h,2h} \right) \mathbf{E}_{m+1}^{1,1} \Bigg] \mathcal{P}_{\mathrm{IRL}}^{\dagger}, \quad (91)
$$

where $\mathcal{P}_{\mathrm{IRL}}$ projects the Hilbert space $(\mathbb{C}^3)^{\otimes L}$ into the constraint Hilbert space for the IRL model. The local operators $\mathbf{E}_m^{a,b}$ are defined in the same way as in the 3-state AFM Hamiltonian.

Since we gauge the $\mathbb{Z}_2$ symmetry (84) of the 3-state AFM model, which is Abelian. We find the twisted Hamiltonian using the method outlined in [13]. The twisted 3-state AFM model is defined in a different constrained Hilbert space, $\mathscr{H}_{\mathbb{Z}_3}^{(b)} \subset (\mathbb{C}^3)^L$, where the state $|s_1, s_2, \ldots, s_L\rangle$ satisfies

$$
s_j \neq s_{j+1}, \quad s_j \in \{0,1,2\}, \quad j \in \{1,2,\ldots,L-1\}; \quad \bar{s}_L \neq \bar{s}_1, \quad (92)
$$

where

$$
\bar{s} := \begin{cases} 0, & s = 0, \\ -s, & s \neq 0. \end{cases} \quad (93)
$$

The dimension of the Hilbert space of the twisted 3-state AFM model is $2^L$ [13]. The twisted 3-state AFM Hamiltonian becomes

$$
\begin{aligned}
\mathbf{H}_{\mathbb{Z}_3,(b)} = \mathcal{P}_{\mathbb{Z}_3,b} \Bigg[ & \sum_{m=1}^{L-2} \sum_{s=0}^{2} \mathbf{E}_{m-1}^{s,s} \mathbf{E}_{m+1}^{s,s} \left( \mathbf{E}_{m}^{s-1,s+1} + \mathbf{E}_{m}^{s+1,s-1} \right) \\
& + \Delta \left( \mathbf{E}_{m-1}^{s,s} \mathbf{E}_{m+1}^{s+1,s+1} + \mathbf{E}_{m-1}^{s,s} \mathbf{E}_{m+1}^{s-1,s-1} - \mathbb{I}_{\mathbb{Z}_3} \right) \\
& + \sum_{s=0}^{2} \mathbf{E}_{L-1}^{s,s} \mathbf{E}_{1}^{\bar{s},\bar{s}} \left( \mathbf{E}_{L}^{s-1,s+1} + \mathbf{E}_{L}^{s+1,s-1} \right) \\
& + \Delta \left( \mathbf{E}_{L-1}^{s,s} \mathbf{E}_{1}^{\overline{s+1},\overline{s+1}} + \mathbf{E}_{L-1}^{s,s} \mathbf{E}_{1}^{\overline{s-1},\overline{s-1}} - \mathbb{I}_{\mathbb{Z}_3} \right) \\
& + \mathbf{E}_{L}^{s,s} \mathbf{E}_{2}^{\bar{s},\bar{s}} \left( \mathbf{E}_{1}^{\overline{s-1},\overline{s+1}} + \mathbf{E}_{1}^{\overline{s+1},\overline{s-1}} \right) \\
& + \Delta \left( \mathbf{E}_{L}^{s,s} \mathbf{E}_{2}^{\overline{s+1},\overline{s+1}} + \mathbf{E}_{L}^{s,s} \mathbf{E}_{2}^{\overline{s-1},\overline{s-1}} - \mathbb{I}_{\mathbb{Z}_3} \right) \Bigg] \mathcal{P}_{\mathbb{Z}_3,b}^{\dagger},
\end{aligned} \quad (94)
$$

where $\mathcal{P}_{\mathbb{Z}_3,b} \cdots \mathcal{P}_{\mathbb{Z}_3,b}^{\dagger}$ projects the Hilbert space $(\mathbb{C}^3)^L$ to $\mathscr{H}_{\mathbb{Z}_3}^{(b)}$ following the rule in (92).

The duality operator is defined in [13], and the MPO representation is presented in App. D. We have

$$
\mathcal{D}_{\mathbb{Z}_3,\mathrm{IRL}} \cdot \mathbf{H}_{\mathbb{Z}_3} = \mathbf{H}_{\mathrm{IRL}} \cdot \mathcal{D}_{\mathbb{Z}_3,\mathrm{IRL}}, \quad (95)
$$

$$
\mathcal{D}_{\mathbb{Z}_3,\mathrm{IRL}}^{(b)} \cdot \mathbf{H}_{\mathbb{Z}_3,(b)} = \mathbf{H}_{\mathrm{IRL}} \cdot \mathcal{D}_{\mathbb{Z}_3,\mathrm{IRL}}^{(b)}. \quad (96)
$$

The symmetry of the IRL model will be explained in Sec. 4.4, when we combine two gauging procedures.

## 4.3 Ising zig-zag to integrable Rydberg ladder

The IRL Hamiltonian can be obtained via duality from the Izz Hamiltonian. In this scenario, we gauge one Frobenius subalgebra of Rep($S_3$) category discussed in Sec. 2.2,[14] instead of discrete

---

[14]A thorough discussion on the Frobenius subalgebra of Rep($S_3$) category can be found in Sec. 3.1.4 of [20]. For a sample of recent discussions of gauging non-invertible symmetries, see [20,21].

Table 4: Character table of the $S_3$ group. $\{1, r, s\}$ are the simple objects in Abelian category $\text{Rep}(S_3)$, and $[1], [a], [b]$ are the conjugacy classes of $S_3$.

|   | $[1]$ | $[a]$ | $[b]$ |
|---|---|---|---|
| $1$ | $1$ | $1$ | $1$ |
| $r$ | $1$ | $1$ | $-1$ |
| $s$ | $2$ | $-1$ | $0$ |

groups in the previous cases. The duality operator $\mathcal{D}_{\text{Izz,IRL}}$ is weakly symmetric, for which the discussion of the Hilbert space decomposition and the twist Hamiltonians will be postponed to future work. However, in the next section, we will use the duality operator $\mathcal{D}_{\text{Izz,IRL}}$ to study the gauging of a non-Abelian discrete group $S_3$. Hence, we outline a few minimal properties of the duality operator and reserve further detailed discussions for future work.

The duality operator maps the Izz Hamiltonian to the IRL Hamiltonian,

$$\mathcal{D}_{\text{Izz,IRL}} \cdot \mathbf{H}_{\text{Izz}} = \mathbf{H}_{\text{IRL}} \cdot \mathcal{D}_{\text{Izz,IRL}}, \tag{97}$$

where both Hamiltonians are defined in (9) and (91). The duality operator can be written as an MPO, where the details can be found in App. E.

As shown in [13], the duality operator satisfies

$$\begin{aligned} \mathcal{D}^{\dagger}_{\text{Izz,IRL}} \cdot \mathcal{D}_{\text{Izz,IRL}} &= \mathbb{I} + \mathbf{S}, \\ \mathcal{D}_{\text{Izz,IRL}} \cdot \mathcal{D}^{\dagger}_{\text{Izz,IRL}} &= \mathbb{I}_{\text{IRL}} + \mathbf{S}_{\text{IRL}}, \end{aligned} \tag{98}$$

where $\mathbf{S} = \mathcal{O}_Z(2\pi/3)$ and $\mathbf{S}_{\text{IRL}}$ are the representations of the non-invertible object of the $\text{Rep}(S_3)$ category in Hilbert spaces $\mathcal{H}_{\text{Izz}}$ and $\mathcal{H}_{\text{IRL}}$, respectively. We can see from the properties of the duality operators (98) that we are gauging the Frobenius subalgebra $\mathcal{A} = (1+s)$.[15] These relations (98) is compatible with the fact that the duality operator $\mathcal{D}_{\text{Izz,IRL}}$ is not strongly symmetric, since

$$\mathcal{D}_{\text{Izz,IRL}} \neq \mathcal{D}_{\text{Izz,IRL}} \cdot \mathbf{R}, \qquad \mathcal{D}_{\text{Izz,IRL}} \cdot (\mathbb{I} + \mathbf{R}) = \mathcal{D}_{\text{Izz,IRL}} \cdot \mathbf{S}. \tag{99}$$

## 4.4 XXZ to integrable Rydberg latter: Gauging the $S_3$ symmetry

Now we move on to the gauging of a non-Abelian discrete group symmetry. We concentrate on the example of gauging the $S_3$ symmetry of the XXZ model.

The $S_3$ group has six group elements $\{1, a, a^2, b, a \cdot b, a^2 \cdot b\}$, with multiplication rules

$$a^3 = b^2 = (a^s \cdot b)^2 = 1, \qquad a^s b = b a^{3-s}, \quad s \in \{0, 1, 2\}. \tag{100}$$

The irreducible representations of $S_3$ are characterized by the $\text{Rep}(S_3)$ category, with simple objects $\{1, r, s\}$. The fusion algebra is given in (31).

The $S_3 = \mathbb{Z}_3 \rtimes \mathbb{Z}_2 \subset SO(2) \rtimes \mathbb{Z}_2 = O(2)$ is a discrete symmetry of the XXZ model.

**Duality operators and twist sectors** The dual theory is the IRL model, and the duality operator can be obtained by decomposing the gauging into two separate gaugings: first the $\mathbb{Z}^z_3$ symmetry of the XXZ chain then the $\mathbb{Z}_2$ symmetry of the 3-state AFM model (see Fig. 2). Interestingly, as shown in [13], we obtain the same gauging by gauging first the $\mathbb{Z}^x_2$ symmetry

---

[15]For a similar discussion in the field theory context, see Sec. 4.1.1 of [20].

of the XXZ model, and then the Rep($S_3$) categorical symmetry of the Izz model (see Fig. 2), i.e.

$$\mathcal{D}_{\text{IRL}} = \mathcal{D}_{\mathbb{Z}_3,\text{IRL}} \cdot \mathcal{D}_{\mathbb{Z}_3} = \mathcal{D}_{\text{Izz,IRL}} \cdot \mathcal{D}_{\text{Izz}} . \tag{101}$$

It is straightforward to observe that the duality operator transforms the XXZ model to the IRL model, i.e.

$$\mathcal{D}_{\text{IRL}} \cdot \mathbf{H}_{\text{XXZ}} = \mathbf{H}_{\text{IRL}} \cdot \mathcal{D}_{\text{IRL}} . \tag{102}$$

From decomposing the duality operator into two separate parts, we obtain the duality operators between the twisted XXZ Hamiltonian and the IRL Hamiltonian,

$$\mathcal{D}_{\text{IRL}}^{(a)} \cdot \mathbf{H}_{\text{XXZ},(\omega,Z)} = \mathbf{H}_{\text{IRL}} \cdot \mathcal{D}_{\text{IRL}}^{(a)} , \tag{103}$$

$$\mathcal{D}_{\text{IRL}}^{(b)} \cdot \mathbf{H}_{\text{XXZ},(-1,X)} = \mathbf{H}_{\text{IRL}} \cdot \mathcal{D}_{\text{IRL}}^{(b)} , \tag{104}$$

where the duality operators are decomposed as

$$\mathcal{D}_{\text{IRL}}^{(a)} = \mathcal{D}_{\mathbb{Z}_3,\text{IRL}} \cdot \mathcal{D}_{\mathbb{Z}_3}^{(\omega)} , \tag{105}$$

$$\mathcal{D}_{\text{IRL}}^{(b)} = \mathcal{D}_{\text{Izz,IRL}} \cdot \mathcal{D}_{\text{Izz}}^{(-1)} . \tag{106}$$

These three twists cover the three conjugacy classes of $S_3$, i.e.

$$[1] = \{1\} , \qquad [a] = \{a, a^2\} , \qquad [b] = \{b, ab, a^2 b\} . \tag{107}$$

We have checked the conjectures in Sec. 3.1, and they are satisfied given the character table of $S_3$ in Table 4,

$$\begin{aligned} \mathcal{D}_{\text{IRL}} \cdot \mathcal{D}_{\text{IRL}}^{\dagger} &= \mathbb{I}_{\text{IRL}} + \mathbf{R}_{\text{IRL}} + 2\,\mathbf{S}_{\text{IRL}} , \\ \mathcal{D}_{\text{IRL}}^{(a)} \cdot \left( \mathcal{D}_{\text{IRL}}^{(a)} \right)^{\dagger} &= \mathbb{I}_{\text{IRL}} + \mathbf{R}_{\text{IRL}} - \mathbf{S}_{\text{IRL}} . \\ \mathcal{D}_{\text{IRL}}^{(b)} \cdot \left( \mathcal{D}_{\text{IRL}}^{(b)} \right)^{\dagger} &= \mathbb{I}_{\text{IRL}} - \mathbf{R}_{\text{IRL}} . \end{aligned} \tag{108}$$

Moreover, by adding up all the products of duality operators, we have

$$\sum_{g \in S_3} \mathcal{D}_{\text{IRL}}^{(g)} \cdot \left( \mathcal{D}_{\text{IRL}}^{(g)} \right)^{\dagger} = \mathcal{D}_{\text{IRL}} \cdot \mathcal{D}_{\text{IRL}}^{\dagger} + 2\mathcal{D}_{\text{IRL}}^{(a)} \cdot \left( \mathcal{D}_{\text{IRL}}^{(a)} \right)^{\dagger} + 3\mathcal{D}_{\text{IRL}}^{(b)} \cdot \left( \mathcal{D}_{\text{IRL}}^{(b)} \right)^{\dagger} = 6\,\mathbb{I}_{\text{IRL}} , \tag{109}$$

where the order of group $S_3$ is precisely 6, matching up with the conjecture (50).

A remark is in order. Suppose that we do not assume the Rep($S_3$) symmetry of the IRL model. From (108), we have three projectors that commute with the IRL Hamiltonian (91). Taking the linear combination of the three projectors in (108), we obtain the operators with Rep($S_3$) fusion algebra, i.e.

$$\begin{aligned} \mathbf{S}_{\text{IRL}} &= \frac{1}{3} \left( \mathcal{D}_{\text{IRL}} \cdot \mathcal{D}_{\text{IRL}}^{\dagger} + \mathcal{D}_{\text{IRL}}^{(a)} \cdot \left( \mathcal{D}_{\text{IRL}}^{(a)} \right)^{\dagger} \right) , \\ \mathbf{R}_{\text{IRL}} &= \frac{1}{6} \left( \mathcal{D}_{\text{IRL}} \cdot \mathcal{D}_{\text{IRL}}^{\dagger} + 2\mathcal{D}_{\text{IRL}}^{(a)} \cdot \left( \mathcal{D}_{\text{IRL}}^{(a)} \right)^{\dagger} - 3\mathcal{D}_{\text{IRL}}^{(b)} \cdot \left( \mathcal{D}_{\text{IRL}}^{(b)} \right)^{\dagger} \right) , \end{aligned} \tag{110}$$

which also commute with the IRL Hamiltonian (91). This can be considered as a demonstration of the assumption that the dual model has the Rep($G$) symmetry after gauging the $G$ symmetry of the original model.

The Rep($S_3$) operators are $\mathbf{R}_{\text{IRL}}$ and $\mathbf{S}_{\text{IRL}}$, where the fusion algebra becomes

$$\mathbf{R}_{\text{IRL}}^2 = \mathbb{I}_{\text{IRL}} , \qquad \mathbf{R}_{\text{IRL}} \cdot \mathbf{S}_{\text{IRL}} = \mathbf{S}_{\text{IRL}} , \qquad \mathbf{S}_{\text{IRL}}^2 = \mathbb{I}_{\text{IRL}} + \mathbf{R}_{\text{IRL}} + \mathbf{S}_{\text{IRL}} . \tag{111}$$

The $\mathbb{Z}_2$ operator reads

$$\mathbf{R}_{\text{IRL}} = \mathcal{P}_{\text{IRL}} \left( \prod_{n=1}^{L} \begin{pmatrix} 0 & 0 & 1 \\ 0 & 1 & 0 \\ 1 & 0 & 0 \end{pmatrix}_n \right) \mathcal{P}_{\text{IRL}}^{\dagger} . \tag{112}$$

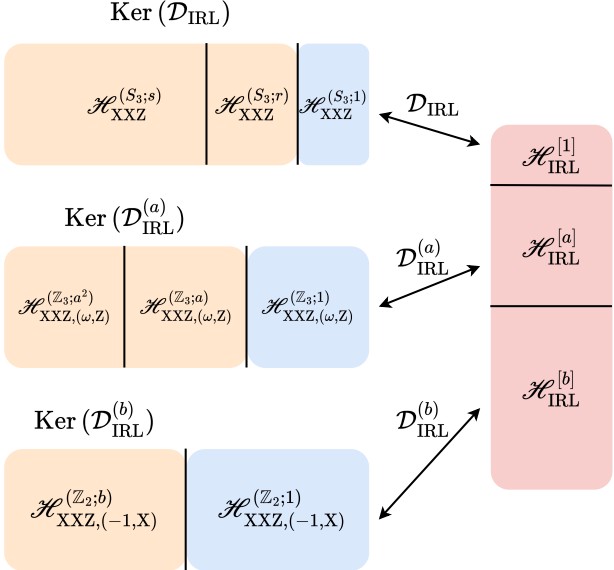

Figure 4: The mapping between the Hilbert spaces of twisted XXZ models and the IRL model.

**Symmetries** The global symmetries of the IRL model within each sector can be obtained as the ring of double cosets from the symmetry of (un)twisted XXZ Hamiltonian, using the method outlined in Sec. 3, i.e.

$$
\begin{aligned}
\mathcal{H}_{\mathrm{IRL}}^{[1]}: \quad & \mathbb{Z}[S_3\backslash O(2)/S_3] \simeq \mathbb{Z}[\mathbb{Z}_2\backslash O(2)/\mathbb{Z}_2], \\
\mathcal{H}_{\mathrm{IRL}}^{[a]}: \quad & \frac{U(1)^z}{\mathbb{Z}_3^z} \simeq U(1), \\
\mathcal{H}_{\mathrm{IRL}}^{[b]}: \quad & \frac{\mathbb{Z}_2^z \times \mathbb{Z}_2^x}{\mathbb{Z}_2^x} \simeq \mathbb{Z}_2,
\end{aligned}
\tag{113}
$$

where we assume that the number of sites $L$ is even.

First of all, we again find the cosine symmetry in the $\mathcal{H}_{\mathrm{IRL}}^{[1]}$ sector:

$$
\mathcal{O}_{\cos}^{\mathrm{IRL}}(\phi) = \frac{1}{6}\mathcal{D}_{\mathrm{IRL}}^{(1)} \cdot \mathcal{O}_z(\phi) \cdot \left(\mathcal{D}_{\mathrm{IRL}}^{(1)}\right)^{\dagger},
\tag{114}
$$

where

$$
\mathcal{O}_{\cos}^{\mathrm{IRL}}(\phi) \cdot \mathcal{O}_{\cos}^{\mathrm{IRL}}(\theta) = \frac{1}{2}\left[\mathcal{O}_{\cos}^{\mathrm{IRL}}(\phi+\theta) + \mathcal{O}_{\cos}^{\mathrm{IRL}}(\phi-\theta)\right].
\tag{115}
$$

From the perspective of group theory, the ring of double cosets $\mathbb{Z}[S_3\backslash O(2)/S_3] \simeq \mathbb{Z}[\mathbb{Z}_2\backslash O(2)/\mathbb{Z}_2]$ gives rise to the same cosine symmetry as the Izz case in 2.

What is more interesting is that in the $\mathcal{H}_{\mathrm{IRL}}^{[a]}$ sector, we have a hidden $U(1)$ symmetry:

$$
\mathcal{O}_{U(1)}^{\mathrm{IRL}}(\phi) = \frac{1}{3}\mathcal{D}_{\mathrm{IRL}}^{(a)} \cdot \mathcal{O}_z(\phi) \cdot \left(\mathcal{D}_{\mathrm{IRL}}^{(a)}\right)^{\dagger},
\tag{116}
$$

where

$$
\mathcal{O}_{U(1)}^{\mathrm{IRL}}(\phi) \cdot \mathcal{O}_{U(1)}^{\mathrm{IRL}}(\theta) = \mathcal{O}_{U(1)}^{\mathrm{IRL}}(\phi+\theta).
\tag{117}
$$

We can further write down the $U(1)$ generator of this symmetry, which is non local,

$$
-\mathrm{i}\frac{\mathrm{d}}{\mathrm{d}\phi}\mathcal{O}_{U(1)}^{\mathrm{IRL}}(\phi)\bigg|_{\phi=0} = \sum_{j=1}^{L}\frac{1}{3}\mathcal{D}_{\mathrm{IRL}}^{(a)} \cdot \sigma_j^z \cdot \left(\mathcal{D}_{\mathrm{IRL}}^{(a)}\right)^{\dagger}.
\tag{118}
$$

This $U(1)$ symmetry has not been mentioned in the previous literature on the IRL model, and it can be obtained in a straightforward manner from the sandwiched construction. However, it is not possible to express the $U(1)$ density in terms of local operators, at least not through the sandwiched construction.

In the $\mathscr{H}_{\text{IRL}}^{[b]}$ sector, the remaining symmetry after the duality transformation is the $\mathbb{Z}_2$ symmetry, i.e.

$$\frac{1}{2}\mathcal{D}_{\text{IRL}}^{(b)}\prod_{n=1}^{L} Z_n\left(\mathcal{D}_{\text{IRL}}^{(b)}\right)^{\dagger} = \frac{1}{2}\left(\mathbb{I}_{\text{IRL}} - \mathbf{R}_{\text{IRL}}\right)\mathsf{F}_{\text{IRL}}, \tag{119}$$

where the $\mathbb{Z}_2$ generator is

$$\mathsf{F}_{\text{IRL}} = \mathcal{P}_{\text{IRL}}\left(\prod_{n=1}^{L/2}\begin{pmatrix} 0 & 0 & 1 \\ 0 & 1 & 0 \\ 1 & 0 & 0 \end{pmatrix}_{2n}\right)\mathcal{P}_{\text{IRL}}^{\dagger}, \tag{120}$$

commuting with the $\text{Rep}(S_3)$ generators $\mathbf{R}_{\text{IRL}}$ and $\mathbf{S}_{\text{IRL}}$.

Therefore, the global symmetry of the entire dual Hilbert space $\mathscr{H}_{\text{IRL}}$ becomes

$$\mathbb{Z}_2 \times \text{Rep}(S_3), \tag{121}$$

which implies

$$[\mathsf{F}_{\text{IRL}}, \mathbf{H}_{\text{IRL}}] = [\mathbf{R}_{\text{IRL}}, \mathbf{H}_{\text{IRL}}] = [\mathbf{S}_{\text{IRL}}, \mathbf{H}_{\text{IRL}}] = 0. \tag{122}$$

At the isotropic point, we have $\mathbb{Z}[S_3\backslash SU(2)/S_3]$ symmetry in the $\mathscr{H}_{\text{IRL}}^{[1]}$ sector, where a cosine symmetry $\mathbb{Z}[S_3\backslash O(2)/S_3] \simeq \mathbb{Z}[\mathbb{Z}_2\backslash SU(2)/\mathbb{Z}_2]$ is included.

In the $\mathscr{H}_{\text{IRL}}^{[a]}$ sector, we have the same $U(1)$ symmetry, i.e. the $U(1)^z$ symmetry in $\mathscr{H}_{\text{XXZ}}$ after gauging the $\mathbb{Z}_3^z$ symmetry.

We now move on to the $\mathscr{H}_{\text{IRL}}^{[b]}$ sector. The symmetry of the $(-1, X)$ twisted XXX Hamiltonian is $O(2)' = U(1)^x \rtimes \mathbb{Z}_2^z$, and we gauge the $\mathbb{Z}_2^x$ symmetry, resulting in the symmetry of $\mathscr{H}_{\text{IRL}}^{[b]}$ sector as

$$O(2)^* = \frac{U(1)^x \rtimes \mathbb{Z}_2^z}{\mathbb{Z}_2^x}, \tag{123}$$

where we use $O(2)^*$ to distinguish the $O(2)$ symmetry coming from the ungauged $U(1)^z \rtimes \mathbb{Z}_2^x$ symmetry.

Therefore, the global symmetry of the entire dual Hilbert space $\mathscr{H}_{\text{IRL}}$ at isotropic point is again

$$\mathbb{Z}_2 \times \text{Rep}(S_3), \tag{124}$$

where the $U(1)$ part of the $O(2)^*$ symmetry in the $\mathscr{H}_{\text{IRL}}^{[b]}$ sector is different from the $U(1)$ symmetry in the $\mathscr{H}_{\text{IRL}}^{[a]}$ sector.

**Remark.** The global symmetries of the models away from the isotropic point that are discussed in this section are shared by a whole family of models, which are not related to the quantum integrability structure of the models. To understand this, we decompose the XXZ spin chain Hamiltonian (up to a constant) into projectors, i.e.

$$\mathbf{H}_{\text{XXZ}} = \sum_{j=1}^{L} 2\left(\mathbf{P}_j^{(0)} + \Delta\mathbf{P}_j^{(1)}\right), \tag{125}$$

where the projectors are

$$\mathbf{P}_j^{(0)} = \frac{1}{2}(X_j X_{j+1} + Y_j Y_{j+1}), \qquad \mathbf{P}_j^{(1)} = \frac{1}{2}(\mathbb{I} + Z_j Z_{j+1}). \tag{126}$$

Both local projectors commute with the $O(2)$ generators, and they form a representation of the chromatic algebra with $Q = 3$ [13]. Therefore, we can consider a family of generic models,

$$\mathbf{H}_g = \sum_{j=1}^{L} \prod_{k,\alpha_k} c_{j,k} \mathbf{P}_{j+k}^{(\alpha_k)}, \quad \alpha_k \in \{0,1\}, \tag{127}$$

all of which have the global $O(2)$ symmetry and are generally not integrable.

We then can perform the same duality transformations on any generic Hamiltonian in the form of (127), resulting in the dual models that have global symmetries discussed in this section depending the gauged symmetries, including the non-invertible ones. Therefore, our method outlined here is rather versatile in this regard.

## 5 Conclusion

In this article, we discussed how global symmetries transform under a non-invertible duality transformation which corresponds to gauging a discrete group symmetry in (1+1)-D quantum lattice models. Despite numerous previous studies in a similar setup, a thorough explanation of global symmetries of the dual model is still absent. We aim to take steps to fill this gap.

We start with strongly symmetric dualities that physically imply the gauging of a finite (non-)Abelian discrete group. We make a few conjectures about the properties of duality operators. By including the contribution of the twisted Hamiltonians, the global symmetries within each subsector of the dual Hilbert space can be obtained with the sandwiched structure.

We then exemplify the general theory using the XXZ model and its duals by gauging the discrete symmetries $\mathbb{Z}_2^x$, $\mathbb{Z}_3^z$ and $S_3$. The global symmetry $O(2)$ ($SU(2)$ in the isotropic case) gets transformed according to the sandwiched construction.

In the example of the Izz model, we present a derivation of the "cosine symmetry" and attribute it to the mathematical structure of the algebraic ring of double cosets $\mathbb{Z}[\mathbb{Z}_2 \backslash O(2)/ \mathbb{Z}_2]$.

In the example of the 3-state AFM model, an additional $\mathbb{Z}_2$ symmetry is present, in addition to the global symmetries obtained through dualities. Combining with global symmetries in each sectors of the Hilbert space, we obtain the global symmetry of the 3-state AFM model as $(U(1) \times \mathbb{Z}_3) \rtimes \mathbb{Z}_2$.

As for gauging the non-Abelian $S_3$ symmetry, we find the non-invertible "cosine symmetry" in one of the sectors of the Hilbert space, while a $U(1)$ symmetry is present in another sector, which remained undetected previously. The global symmetry of the entire dual Hilbert space is identified as $\mathbb{Z}_2 \times \mathrm{Rep}(S_3)$.

Similar to the examples in the article, we expect to find the global symmetries of other dual models related to the gauging of discrete group symmetry. However, the current construction needs to be improved to include the scenario with weakly symmetric dualities, which we plan to investigate in future work, making use of the knowledge of the category theory.

Another interesting point is that the models we study, with homogeneous coupling, are integrable. It is possible to construct the transfer matrices that are the generating functions of infinitely many (quasi-)local conserved charges. In the scope of the current article, we discuss how global symmetries (regardless of integrability) transform under the dualities. Equivalently, one might wonder how the integrable structure (e.g. transfer matrices) changes under the duality transformation. We will address this question in future work.

# Acknowledgments

We are grateful to Paul Fendley, Linhao Li, Henry Liu, Kantaro Ohmori, Yuji Tachikawa, Eric Vernier, Xingyang Yu, Yi Zhang, Yunqin Zheng and Yehao Zhou for valuable discussions.

**Funding information** This work was supported by World Premier International Research Center Initiative (WPI), MEXT, Japan. This work was supported in part by the JSPS Grant-in-Aid for Scientific Research (Grant No. 20H05860, 23K17689, 23K25865) [MY], and by JST, Japan (PRESTO Grant No. JPMJPR225A, Moonshot R&D Grant No. JPMJMS2061) [MY]. W.C. also acknowledges support from JSPS KAKENHI grant No. JP19H05810, JP22J21553 and JP22KJ1072 and from Villum Fonden Grant no. VIL60714. The authors of this paper were ordered alphabetically.

# A  Evidence for (47) and (48)

We show that (47) and (48) can be derived from the strongly symmetric condition together with a certain Ansatz. This applies whenever gauging a general fusion category $\mathscr{C}$, which naturally includes the case of a group.

We start with a spherical fusion category $\mathscr{C}$ with finitely many simple objects. The fusion symbol among simple objects is

$$a \cdot b = \sum_{c \in \mathscr{C}} N_{ab}{}^c c, \quad a, b, c \in \mathscr{C}. \tag{A.1}$$

In terms of the quantum dimensions of simple objects, we have

$$\dim(a) \dim(b) = \sum_{c \in \mathscr{C}} N_{ab}{}^c \dim(c). \tag{A.2}$$

The duality operator $\mathcal{D} \in \mathrm{Hom}(\mathscr{H}, \tilde{\mathscr{H}})$, where $\mathscr{H}$ and $\tilde{\mathscr{H}}$ are the Hilbert spaces where the Hamiltonian $\mathbf{H}$ and its dual $\tilde{\mathbf{H}}$ are defined, respectively. Hence, the combination $\mathcal{D}^\dagger \cdot \mathcal{D}$ acts on the Hilbert space of $\mathscr{H}$, i.e.

$$\mathcal{D}^\dagger \cdot \mathcal{D} \in \mathrm{End}(\mathscr{H}), \tag{A.3}$$

and it commutes with the Hamiltonian $\mathbf{H}$ and $\mathscr{C}$ symmetry operators,

$$\left[ \mathcal{D}^\dagger \mathcal{D}, \mathbf{H}_1 \right] = \left[ \mathcal{D}^\dagger \mathcal{D}, \mathcal{O}_a \right] = 0, \quad \forall a \in \mathscr{C}. \tag{A.4}$$

It is natural to conjecture that

$$\mathcal{D}^\dagger \cdot \mathcal{D} = \sum_{a \in \mathscr{C}} \alpha_a \mathcal{O}_a, \tag{A.5}$$

where $\alpha_a$ are constants. From the Ansatz above, we have

$$\mathcal{D}^\dagger \mathcal{D} \mathcal{O}_a = \dim(a) \mathcal{D}^\dagger \mathcal{D} = \sum_{b \in \mathscr{C}} \dim(a) \alpha_b \mathcal{O}_b = \sum_{b \in \mathscr{C}} \sum_{c \in \mathscr{C}} N_{ba}{}^c \alpha_b \mathcal{O}_c, \tag{A.6}$$

so that

$$\dim(a) \alpha_c = \sum_{b \in \mathscr{C}} N_{ba}{}^c \alpha_b. \tag{A.7}$$

Using the property of the fusion symbol,

$$N_{ba}{}^c = N_{a\bar{c}}{}^{\bar{b}}, \tag{A.8}$$

we have

$$\dim(a)\alpha_c = \sum_{\bar{b}\in\mathscr{C}} N_{a\bar{c}}{}^{\bar{b}}\alpha_b\,. \tag{A.9}$$

Here we use the fact that the quantum dimensions of an object $b$ and its dual object $\bar{b}$ are equal if $\mathscr{C}$ is spherical [53]. Thanks to the relation (A.2) we find the solution

$$\alpha_b = \dim(\bar{b}) = \dim(b)\,, \quad \forall b \in \mathscr{C}\,. \tag{A.10}$$

Therefore, (47) are obtained from the strongly symmetric condition (38) with a reasonable Ansatz.

Similar analysis works for the dual case with strongly symmetric duality $\mathcal{D}^\dagger$, with Ansatz

$$\mathcal{D}\cdot\mathcal{D}^\dagger = \sum_{\hat{a}\in\hat{\mathscr{C}}} \beta_{\hat{a}}\tilde{\mathcal{O}}_{\hat{a}}\,, \tag{A.11}$$

where $\hat{\mathscr{C}}$ is the symmetry of the dual Hamiltonian from gauging.

We obtain the constants $\beta_{\hat{a}}$ is proportional to the quantum dimension of the object $\hat{a}$, i.e.

$$\beta_{\hat{a}} = \dim(\hat{a})\,, \quad \forall \hat{a} \in \hat{\mathscr{C}}\,. \tag{A.12}$$

## B  Ring of double cosets

For a group $G$ and its subgroup $K$, the elements of double cosets $K\backslash G/K$ are

$$K\backslash G/K := \{K\cdot g\cdot K | g \in G\}\,, \tag{B.1}$$

where

$$K\cdot g\cdot K := \{k_1\cdot g\cdot k_2 | k_1, k_2 \in K\}\,. \tag{B.2}$$

The double cosets have the property that

$$K\cdot g\cdot K = K\cdot(k_1\cdot g\cdot k_2)\cdot K\,, \quad k_1, k_2 \in K\,. \tag{B.3}$$

For instance, we start with $G_1 = S_3 = \mathbb{Z}_3 \rtimes \mathbb{Z}_2 = \{1, a, a^2, b, ab, ab^2\}$ with $a^3 = b^2 = 1$ and $a^n b = ba^{-n}$. $K_1 = \mathbb{Z}_2 = \{1, b\}$ is a non-normal subgroup, and generates the double coset

$$K_1\backslash G_1/K_1 = \{K_1\cdot 1\cdot K_1, K_1\cdot a\cdot K_1\}\,, \tag{B.4}$$

where

$$\begin{aligned}
K_1\cdot 1\cdot K_1 &= K_1\cdot b\cdot K_1 = \{1, b\}\,, \\
K_1\cdot a\cdot K_1 &= K_1\cdot a^2\cdot K_1 = K_1\cdot ab\cdot K_1 = K_1\cdot a^2 b\cdot K_1 = \{a, a^2, ab, a^2 b\}\,.
\end{aligned} \tag{B.5}$$

In the case of normal subgroup $K_2 = \mathbb{Z}_3 = \{1, a, a^2\}$, we have

$$K_2\backslash G_1/K_2 = \{K_2\cdot 1\cdot K_2, K_2\cdot b\cdot K_2\}\,, \tag{B.6}$$

where

$$\begin{aligned}
K_2\cdot 1\cdot K_2 &= K_2\cdot a\cdot K_2 = K_2\cdot a^2\cdot K_2 = \{1, a, a^2\}\,, \\
K_2\cdot b\cdot K_2 &= K_2\cdot ab\cdot K_2 = K_2\cdot a^2 b\cdot K_2 = \{b, ab, a^2 b\}\,.
\end{aligned} \tag{B.7}$$

For a given group $G$ with a normal subgroup $K$, the double coset (and left/right coset that form sets, which are isomorphic to the set of double coset in this case) form the quotient group $G/K$ (Group multiplication can be defined accordingly.).

In the $G = S_3$, $K' = \mathbb{Z}_3$ example, we have the quotient group $G/K' = \mathbb{Z}_2$, where the group multiplication $*$ is defined as

$$(K \cdot g_1 \cdot K) * (K \cdot g_2 \cdot K) = (K \cdot g_1 g_2 \cdot K). \tag{B.8}$$

However, when the subgroup $K$ is non-normal, we can no longer define a group multiplication. Instead, we can define an algebraic ring over field $\mathbb{Z}$, i.e. $\mathbb{Z}[K\backslash G/K]$, with the addition $+$ defined in the usual way and the multiplication $\star$ such that

$$(K \cdot g_1 \cdot K) \star (K \cdot g_2 \cdot K) = \sum_{k \in K} K \cdot (g_1 k g_2) \cdot K, \quad g_1, g_2 \in G. \tag{B.9}$$

The associativity can be can be shown straightforwardly,

$$[(K \cdot g_1 \cdot K) \star (K \cdot g_2 \cdot K)] \star (K \cdot g_3 \cdot K) = (K \cdot g_1 \cdot K) \star [(K \cdot g_2 \cdot K) \star (K \cdot g_3 \cdot K)], \tag{B.10}$$

using the fact that $K \cdot g \cdot K = K \cdot k_1 g k_2 \cdot K$, for any $k_1, k_2 \in K$.

In addition to the multiplication $\star$ for the set of double cosets, we also need to equip the set with the addition from the integer field $\mathbb{Z}$, i.e. $(K\backslash G/K, \mathbb{Z}, +, \star) := \mathbb{Z}[K\backslash G/K]$, which form an algebraic ring over the field $\mathbb{Z}$.[16] The ring axioms can be checked easily.

Therefore, the sandwiched symmetry operators $\tilde{\mathcal{O}}_g$ in Sec. 3 satisfy the properties of the ring of double coset $\mathbb{Z}[K\backslash G/K]$, i.e. a representation of the ring of double coset on the Hilbert space $\tilde{\mathcal{H}}$.

We would like to mention that the concept of the algebraic ring of double cosets can be generalized [60], which plays a role in the context of non-invertible symmetries of field theory and string theory.

Let us present a few examples of the rings of double cosets.

**1. Gauging the $\mathbb{Z}_2 = \{1, a\}$ symmetry of $O(2)$ symmetry.** We arrive at the ring of double coset $\mathbb{Z}[\mathbb{Z}_2\backslash O(2)/\mathbb{Z}_2]$.

The $U(1) \subset O(2)$ symmetry generators read

$$\phi \cdot \theta = (\phi + \theta), \quad \phi, \theta \in \mathbb{R}. \tag{B.11}$$

The double coset then read

$$\mathbb{Z}_2\backslash O(2)/\mathbb{Z}_2 = \{\mathbb{Z}_2 \cdot \phi \cdot \mathbb{Z}_2 | \phi \geq 0\}, \tag{B.12}$$

where

$$\mathbb{Z}_2 \cdot \phi \cdot \mathbb{Z}_2 = \mathbb{Z}_2 \cdot (-\phi) \cdot \mathbb{Z}_2 = \mathbb{Z}_2 \cdot (\phi \cdot a) \cdot \mathbb{Z}_2. \tag{B.13}$$

The multiplication thus becomes

$$(\mathbb{Z}_2 \cdot \phi \cdot \mathbb{Z}_2) \star (\mathbb{Z}_2 \cdot \theta \cdot \mathbb{Z}_2) = \mathbb{Z}_2 \cdot (\phi + \theta) \cdot \mathbb{Z}_2 + \mathbb{Z}_2 \cdot (\phi - \theta) \cdot \mathbb{Z}_2. \tag{B.14}$$

It is straightforward to check that the cosine symmetry in the Izz model is a representation of the ring of double coset $\mathbb{Z}[\mathbb{Z}_2\backslash O(2)/\mathbb{Z}_2]$ described above.

**2. Gauging $\mathbb{Z}_2^x$ symmetry of $SU(2)$ symmetry.** We arrive at the ring of double coset $\mathbb{Z}[\mathbb{Z}_2\backslash SU(2)/\mathbb{Z}_2]$.

The group element of $SU(2)$ symmetry is denoted as

$$(\phi_x, \phi_y, \phi_z) = \exp((i\phi_x s^x) \exp(i\phi_y s^y) \exp(i\phi_z s^z)). \tag{B.15}$$

We discuss the sub-rings of the ring of double coset. The subrings generated by either $(0, 0, \phi_z)$ or $(0, \phi_y, 0)$ are both isomorphic to the ring of double coset $\mathbb{Z}[\mathbb{Z}_2\backslash O(2)/\mathbb{Z}_2]$. The sub-ring $(\phi_x, 0, 0)$ becomes $\mathbb{Z}[U(1)]$, i.e. the group ring of the quotient group $\dfrac{U(1)^x}{\mathbb{Z}_2^x} = U(1)$.

---

[16]The ring of double cosets defined here is known as the Hecke ring $\mathcal{H}(G, K)$ in the mathematical literature, cf. Chapter V of [59].

**3. Gauging $\mathbb{Z}_3^z$ symmetry of $SU(2)$ symmetry.** We arrive at the ring of double coset $\mathbb{Z}[\mathbb{Z}_3 \backslash SU(2)/\mathbb{Z}_3]$.

The subrings generated by $(\phi_x, 0, 0)$ and $(0, \phi_y, 0)$ satisfy the following multiplication

$$
\begin{aligned}
&\left[\mathbb{Z}_3^z \cdot (\phi_x, 0, 0) \cdot \mathbb{Z}_3^z\right] \star \left[\mathbb{Z}_3^z \cdot (\theta_x, 0, 0) \cdot \mathbb{Z}_3^z\right] \\
&= \left(\mathbb{Z}_3^z \cdot (\phi_x + \theta_x, 0, 0) \cdot \mathbb{Z}_3^z + \mathbb{Z}_3^z \cdot \left[(\phi_x, 0, 0) \cdot \left(0, 0, \frac{2\pi}{3}\right) \cdot (\theta_x, 0, 0)\right] \cdot \mathbb{Z}_3^z \right. \\
&\quad \left. + \mathbb{Z}_3^z \cdot \left[(\phi_x, 0, 0) \cdot \left(0, 0, -\frac{2\pi}{3}\right) \cdot (\theta_x, 0, 0)\right] \cdot \mathbb{Z}_3^z \right).
\end{aligned}
\tag{B.16}
$$

The sub-ring $(0, 0, \phi_z)$ again becomes $\mathbb{Z}[U(1)]$, i.e. the group ring of the quotient group $\frac{U(1)^z}{\mathbb{Z}_3^z} = U(1) \supset \mathbb{Z}[\mathbb{Z}_3 \backslash SU(2)/\mathbb{Z}_3]$.

**4. Gauging $S_3 = \mathbb{Z}_3^z \rtimes \mathbb{Z}_2^x$ symmetry of $SU(2)$ symmetry.** We arrive at the ring of double coset $\mathbb{Z}[S_3 \backslash SU(2)/S_3]$.

For the part $(0, 0, \phi_z)$, we have a sub-ring of double coset $\mathbb{Z}[S_3 \backslash O(2)/S_3] = \mathbb{Z}[\mathbb{Z}_2^x \backslash O(2)/\mathbb{Z}_2^x]$ inside $\mathbb{Z}[S_3 \backslash SU(2)/S_3]$.

$$
\begin{aligned}
&\left[S_3 \cdot (0, 0, \phi_z) \cdot S_3\right] \star \left[S_3 \cdot (0, 0, \theta_z) \cdot S_3\right] \tag{B.17}\\
&= \left(S_3 \cdot (0, 0, \phi_z + \theta_z) \cdot S_3 + S_3 \cdot \left(0, 0, \phi_z + \theta_z + \frac{2\pi}{3}\right) \cdot S_3 + S_3 \cdot \left(0, 0, \phi_z + \theta_z + \frac{4\pi}{3}\right) \cdot S_3 \right. \\
&\quad \left. + S_3 \cdot (0, 0, \phi_z - \theta_z) \cdot S_3 + S_3 \cdot \left(0, 0, \phi_z - \theta_z + \frac{2\pi}{3}\right) \cdot S_3 + S_3 \cdot \left(0, 0, \phi_z - \theta_z + \frac{4\pi}{3}\right) \cdot S_3 \right) \\
&= 3 \left(S_3 \cdot (0, 0, \phi_z + \theta_z) \cdot S_3 + S_3 \cdot (0, 0, \phi_z - \theta_z) \cdot S_3\right),
\end{aligned}
$$

where we use the properties that

$$
S_3 \cdot (0, 0, \phi_z) \cdot S_3 = S_3 \cdot \left(0, 0, \phi_z + \frac{2\pi}{3}\right) \cdot S_3 = S_3 \cdot \left(0, 0, \phi_z + \frac{4\pi}{3}\right) \cdot S_3,
\tag{B.18}
$$

as a consequence of gauging the $\mathbb{Z}_3$ symmetry. The formula is the same as the one of cosine symmetry again.

# C MPO as cosine symmetry

In Sec. 2.2, we show that the $O(2)$ symmetry of the XXZ model becomes the ring of double cosets $\mathbb{Z}[\mathbb{Z}_2 \backslash O(2)/\mathbb{Z}_2]$ in the Izz model. Moreover, the symmetry operators of the Izz model satisfy the so-called cosine rule (21).

The cosine rule can be readily derived from the property of the MPO, i.e.

$$
\begin{aligned}
\mathcal{O}_z^{\text{Izz}}(\phi) &= \mathcal{D}_{\text{Izz}} \left(\prod_{n=1}^{L} \exp\left(i\phi Z_n\right)\right) \mathcal{D}_{\text{Izz}}^{\dagger} \\
&= \text{Tr}_a \left(\prod_{n=1}^{L} \begin{pmatrix} \cos\phi & -\sin\phi X_n \\ -\sin\phi & -\cos\phi X_n \end{pmatrix}_a\right) = \frac{1}{2} \text{Tr}_a \left(\prod_{n=1}^{L} \mathbf{C}_{a,n}(\phi)\right).
\end{aligned}
\tag{C.1}
$$

Note that a priori $\mathcal{O}_{\text{Izz}}^z(\phi)$ should be an MPO with bond dimension 4. However, by block diagonalizing the auxiliary space, we reduce the bond dimension of $\mathcal{O}_{\text{Izz}}^z(\phi)$ to 2, i.e. a conserved charge with non-local density. The symmetry operator $\mathcal{O}_{\text{Izz}}^z(\phi)$ is related to the *semilocal*

charges discussed in [61]. Then we can perform a similar calculation for the cosine rule,

$$\mathcal{O}_z^{\text{Izz}}(\phi) \cdot \mathcal{O}_z^{\text{Izz}}(\theta) = \text{Tr}_{a,b}\left[\prod_{n=1}^{L}\left(\mathbf{C}_{a,n}(\phi)\otimes \mathbb{I}_b\right)\left(\mathbb{I}_a \otimes \mathbf{C}_{b,n}(\theta)\right)\right]$$
$$= \text{Tr}_c\left(\prod_{n=1}^{L}\mathbf{D}_{c,n}(\phi,\theta)\right), \tag{C.2}$$

where

$$\mathbf{U}_c\mathbf{D}_{c,n}(\phi,\theta)\mathbf{U}_c^\dagger = \begin{pmatrix} \cos(\phi+\theta) & -\sin(\phi+\theta)\,X_n & 0 & 0 \\ -\sin(\phi+\theta) & -\cos(\phi+\theta)\,X_n & 0 & 0 \\ 0 & 0 & \cos(\phi-\theta) & -\sin(\phi-\theta)\,X_n \\ 0 & 0 & -\sin(\phi-\theta) & -\cos(\phi-\theta)\,X_n \end{pmatrix}_c, \tag{C.3}$$

which are block diagonal with a "gauge" transformation in the auxiliary space

$$\mathbf{U}_c = \frac{1}{\sqrt{2}}\begin{pmatrix} 1 & 0 & 0 & -1 \\ 0 & 1 & 1 & 0 \\ 1 & 0 & 0 & 1 \\ 0 & -1 & 1 & 0 \end{pmatrix}. \tag{C.4}$$

By taking the partial trace in the auxiliary space $c$, we readily obtain

$$\mathcal{O}_z^{\text{Izz}}(\phi)\cdot\mathcal{O}_z^{\text{Izz}}(\theta) = \text{Tr}_a\left(\prod_{n=1}^{L}\mathbf{C}_{a,n}(\phi+\theta)\right) + \text{Tr}_b\left(\prod_{n=1}^{L}\mathbf{C}_{b,n}(\phi-\theta)\right)$$
$$= \mathcal{O}_{\text{Izz}}^z(\phi+\theta) + \mathcal{O}_{\text{Izz}}^z(\phi-\theta), \tag{C.5}$$

providing an alternative derivation of the cosine rule in terms of MPOs, cf. (21).

## D  MPO for duality between 3-state AFM model and IRL model

The duality operator between the 3-state AFM model and the IRL model can be written as an MPO [13],

$$\mathcal{D}_{\mathbb{Z}_3,\text{IRL}} = \quad \cdots, \tag{D.1}$$

where $\{s_1, s_2, \ldots, s_L\}$ and $\{h_1, h_2, \ldots, h_L\}$ are states of the 3-state AFM and the IRL model, respectively. The variables $s_i$ and $h_i$ take the value $\{0, 1, 2\}$, while satisfying different constraints of the corresponding Hilbert space, cf. (72) and (90). The non-zero matrix elements are

$$= \quad = 1, \tag{D.2}$$

$$= \quad = 2^{-1/4}(-1)^{\delta_{s,2}\delta_{h,2}}.$$

In the presence of the $\mathbb{Z}_2$ twist, the duality operator can be obtained by inserting a "defect" between the last and the first sites in the MPO, i.e.

$$\mathcal{D}^{(-1)}_{\mathbb{Z}_3,\text{IRL}} = \qquad \cdots , \qquad (\text{D.3})$$

where the "defect" is obtained by adding the $\mathbb{Z}_2$ flip of the first site,

$$\qquad = \qquad . \qquad (\text{D.4})$$

In the presence of twist, the constraint between $s_L$ and $s_1$ changes into $\bar{s}_L \neq s_1$ [13].

## E MPO for duality between Izz model and IRL model

The duality operator between the Izz model and the IRL model can be written as an MPO,

$$\mathcal{D}_{\text{Izz,IRL}} = \qquad \cdots , \qquad (\text{E.1})$$

where $\{j_1, j_2, \ldots, j_L\}$ and $\{h_1, h_2, \ldots, h_L\}$ are states of the Izz and the IRL model, respectively. The variables $j_i \in \{\uparrow, \downarrow\}$ stand for spin-1/2 states. The variables $h_i$ take the values in $\{0, 1, 2\}$ while satisfying the constraints (90) for the IRL Hilbert space [13].

The non-zero matrix elements are

$$= 2^{-1/2} ; \qquad = 2^{-1/2} ; \qquad (\text{E.2})$$

$$= \qquad = 2^{-1/4} ,$$

where $j, j' \in \{\uparrow, \downarrow\}$, the spin-flipped state $|\bar{j}\rangle = X|j\rangle$, and $h \in \{0, 2\}$.

We do not study the twist duality operator in the scope of this article.

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
