# Peer review of "Global symmetries of quantum lattice models under non-invertible dualities"

_SciPost Physics, doi:SciPost Phys. Core 8, 070 (2025)_

## Round 1 · Referee Report · Anonymous (Referee 3) · 2025-9-2

Report

We thank the authors for their detailed response to our list of questions. The updated manuscript addresses all of them. There is one typo in the newly added Table 1. In the second row of the rightmost column, the authors probably meant to write U(1)^x rather than SU(2)^x.

Recommendation

Publish (easily meets expectations and criteria for this Journal; among top 50%)

---

## Round 1 · Author Response

We thank the referees for their careful reading and evaluation of our work. We have improved our draft according to the comments of the referees, and we would like to present some responses to the referee reports below.

Referee Report #1

1-- We thank the referee for the question. The method we developed should work for the gauging of $0$-form symmetry in arbitrary dimensions. We focus on the $(1+1)$-dimensional systems, where the dual symmetry is also a $0$-form symmetry.

2-- We thank the referee for pointing out the mistake. Indeed, what we are dealing with is beyond the scope of fusion category (which has only finite number of simple objects). We changed it into monoidal category to avoid inaccuracy in the draft.

3-- We thank the referee for pointing out this. The strongly symmetric duality appears when gauging a finite discrete group. We also added two footnotes in Sec. 3.1 to stress this.

4-- We thank the referee for pointing out this. Indeed, the decomposition of the Hilbert space of the twisted Hamiltonian is not due to the gauging. It is a simple consequence from the representation theory of finite groups. We corrected the statement by removing the improper part.

5-- We thank the referee for the question. We use the analogous definition of the gauging of Frobenius algebra object in [17]. From the definition, we expect that

$\mathcal{D}_{\rm Izz,IRL}^\dagger \mathcal{D}_{\rm Izz,IRL} = \mathbb{I} + \mathbf{S} \; , $

which signifies the gauging of $\mathcal{A} = 1 + s$. For a more concrete field theoretical approach, we refer to Sec. 4.1.1 of [20], where there is a duality operator that satisfies the same relation as our $\mathcal{D}_{\rm Izz,IRL}$.

6-- We thank the referee for the question. We added an additional equation below Eq. (117) to express the $U(1)$ generator. Since the duality operators $\mathcal{D}_{\rm IRL}^{(a)}$ and $\left( \mathcal{D}_{\rm IRL}^{(a)} \right)^\dagger$ are MPOs of bond dimension 9, we expect the $U(1)$ generator to be at most an MPO of bond dimension 81, which is cumbersome to express explicitly. We hope that the new Eq. (118) would suffice for the request of the referee.

7-- We thank the referee for pointing out the issue. First, in Eq. (47), we were working on the group case, where $\mathrm{dim}(b) = 1$. Second, we made a typo in Eq. (134) as the referee noticed, which should read $\mathcal{D} \cdot \mathcal{D}^\dagger$. We corrected it in the new version.

8-- We thank the referee for pointing out the typo. We corrected Eq. (18).

9-- We thank the referee for pointing out the typo. We corrected Eq. (94).

10-- We thank the referee for pointing out the typo. We corrected Eq. (144).

Referee Report #2

We thank the referee for a careful reading of our draft and for raising some criticisms. Unfortunately, we do not fully agree with the referee's comments, which we hope to explain as follows.

1-- We thank the referee for noticing the definition and properties of the algebraic ring of double cosets. We put them in the appendices because we think that the definition of the double cosets is standard in the group theory literature. Moreover, the algebraic ring of double cosets is also well studied in the field of Hecke algebra. Therefore, we put them in the appendices, since they do not contain any new results and we merely use them as a mathematical tool to describe the physical phenomena.

2-- We thank the referee for the comment. We put some technical details in the appendices to allow the readers to comprehend the main results easier in the main text.

3-- We thank the referee for the comment. We disagree with the referee that "the bulk of the paper is devoted to a single set of examples that have been already done to death in the literature". The global symmetries of dual models of the XXX/XXZ models have not been studied thoroughly in the previous papers of [11-13], though some pieces are presented. These results are collected in Table 2 of the current version, and we could not find a similar result in the previous papers, especially of the isotropic limit. Those examples serve as evidence over the conjectures, and actually our method goes beyond just 4 models, as in the remark at the end of Sec. 4 that we added to the current version. We believe that the examples we present are already strong evidence of the conjecture.

Referee Report #3

1-- We thank the referee for pointing out the typo. We corrected it.

2-- We thank the referee for pointing it out. We change Eq. (2) to $\phi \in [0, \pi)$ to avoid confusion.

3-- We thank the referee for pointing out the issue. We fixed the definition of the algebraic ring of double cosets with the integer field $\mathbb{Z}[...]$. The correct notation is applied to the rest of the draft too.

4-- We thank the referee for the suggestion. We added Table 1 in the beginning of Sec. 4 to collect the global symmetries of the XXX/XXZ models with/without twists.

5-- Yes, the referee's comment is correct here.

6-- We thank the referee for pointing it out. It is related to the comment above, which we fixed the notation throughout the draft now.

7-- We thank the referee for the question. However, we could not find the minus sign in Eq. (34).

8-- We thank the referee for pointing out the typo. In fact, the ``non-Abelian'' here should be ``non-invertible''. We corrected the typos in both places.

9-- We thank the referee for the question. The superscript should not be understood as ``charge'' but as ``conjugacy class''. This is due to the Tannaka dual of the $\mathrm{Rep}(G)$ being the conjugacy class of $G$, which leads to the Hilbert space decomposition.

10-- We thank the referee for the question. It is straightforward to check that the operator $\tilde{\mathcal{O}}_s$ satisfies the two defining properties of the ring of double cosets (144) and (145), therefore it is sufficient to call it a representation of the ring of double cosets.

11-- We thank the referee for pointing out the typo. The referee was correct and we change the notation in Eq. (67) to $|G_g|$ to keep the consistency with the rest of the article.

12-- We thank the referee for pointing out the typo. It was a typo and we corrected it to $\mathbb{Z}[\mathbb{Z}_3 \backslash SU(2) \slash \mathbb{Z}_3]$.

13-- We thank the referee for pointing out the typo. We corrected the typo in Eq. (90).

14-- We thank the referee for the question. On the contrary, the $U(1)$ generator of the IRL model is written in terms of a sum of MPOs. We added an additional equation as the new Eq. (118). The fact that it is a sum of MPOs does not contract with the non-locality. In fact, a generic MPO cannot be written as an operator with local density.

---

## Round 1 · List of Changes

1-- We added two footnotes in Sec. 3.1 about strongly/weakly symmetric dualities.

2-- We added "finite" in the conclusion part, i.e. " the gauging of a finite (non-)Abelian discrete group" to make it more rigorous.

3-- We removed the "a consequence of gauging $G_g$ symmetry" after Eq. (51).

4-- We corrected a typo in Eq. (134).

5-- We corrected a typo in Eq. (18).

6-- We corrected a typo in Eq. (144).

7-- We corrected a typo in page 2 "down-to-earth".

8-- We change Eq. (2) to $\phi \in [0,\pi)$.

9-- We corrected the typos below Eq. (35) and (98). "non-invertible object".

10-- We changed the notation in Eq. (67) to $|G_g|$.

11-- We corrected the typo in the last sentence of Sec. 4.1.

12-- We corrected the typo in Eq. (90).

13-- We corrected the typo in Eq. (94).

14-- We added a table of global symmetries of XXX/XXZ models with/without twists in Table 1, at the beginning of Sec. 4.

15-- We added a remark at the end of Sec. 4 to explain that our method can be easily generalized to the non-integrable cases, so long as the global symmetries remain the same.

---

## Editorial Decision

published